# Cost-aware Stopping for Bayesian Optimization

## Abstract

In automated machine learning, scientific discovery, and other applications of Bayesian optimization, deciding when to stop evaluating expensive black-box functions in a cost-aware manner is an important but underexplored practical consideration. A natural performance metric for this purpose is the *cost-adjusted simple regret*, which captures the trade-off between solution quality and cumulative evaluation cost. While several heuristic or adaptive stopping rules have been proposed, they lack guarantees ensuring stopping before incurring excessive function evaluation costs. We propose a *principled cost-aware stopping rule* for Bayesian optimization that adapts to varying evaluation costs without heuristic tuning. Our rule is grounded in a theoretical connection to state-of-the-art cost-aware acquisition functions, namely the Pandora's Box Gittins Index (PBGI) and log expected improvement per cost (LogEIPC). We prove a theoretical guarantee bounding the expected cost-adjusted simple regret incurred by our stopping rule when paired with either acquisition function. Across synthetic and empirical tasks, including hyperparameter optimization and neural architecture size search, pairing our stopping rule with PBGI or LogEIPC usually matches or outperforms other acquisition-function–stopping-rule pairs in terms of cost-adjusted simple regret.

## 1 Introduction

Bayesian optimization is a framework designed to efficiently find approximate solutions to optimization problems involving expensive-to-evaluate black-box functions, where derivatives are unavailable. Such problems arise in applications like hyperparameter tuning (Snoek et al., 2012), robot control optimization (Martinez-Cantin, 2017), and material design (Zhang et al., 2020). It works by iteratively, (a) forming a probabilistic model of the black-box objective function based on data collected thus far, then (b) optimizing an acquisition function, which balances exploration-exploitation tradeoffs, to carefully choose a new point at which to observe the unknown function in the next iteration.

In this work, we consider the *cost-aware* setting, where one must pay a cost to collect each data point, and study *adaptive stopping rules* that choose when to stop the optimization process. After stopping at some terminal time, we measure performance in terms of simple regret, which is the difference in value between the best solution found so far and the global optimum. Collecting a data point can reduce simple regret, but incurs cost in order to do so.

As an example, consider using a cloud computing environment to tune the hyperparameters of a classifier in order to optimize a performance metric on a given test set. Training and evaluating test error takes some number of CPU or GPU hours, that may depend on the hyperparamaters used. These come with a financial cost, billed by the cloud computing provider, which define our cost function. The objective value is the business value of deploying the trained model under the given hyperparameters—a given function of the model's accuracy. From this perspective, simple regret can be understood as the opportunity cost for deploying a suboptimal model instead of the optimal one. Motivated by the need to balance cost-performance tradeoff in examples such as above, we aim to design stopping rules that optimizes *expected cost-adjusted simple regret*, defined as the sum of simple regret and the cumulative cost of data collection.

Several stopping rules have been proposed for Bayesian optimization. Simple heuristics—such as fixing a maximum number of iterations or stopping when the best value remains unchanged for a certain number of iterations—are widely used in practice, but can either stop too early or lead to unnecessary evaluations. Other approaches include acquisition-function-based rules that stop

when, for instance, the probability of improvement, expected improvement, or knowledge gradient drop below a preset threshold (Lorenz et al., 2015; Nguyen et al., 2017; Frazier & Powell, 2008); and regret-bound-based rules, which use theoretical performance guarantees to decide termination (Makarova et al., 2022; Ishibashi et al., 2023; Wilson, 2024). Most of these target the classical setting which does not explicitly incorporate function evaluation costs.

In this work, we study how to design cost-aware stopping rules, motivated by two primary factors. First, state-of-the-art cost-aware acquisition functions such as the Pandora's Box Gittins Index (PBGI) (Xie et al., 2024) and log expected improvement per cost (LogEIPC) (Ament et al., 2023) have not yet been studied in the adaptive stopping setting. This is important because—as our experiments in Section 4 will show—for best performance, one should pair different acquisition functions with different stopping rules. Second, while certain stopping rules, such as UCB–LCB (Makarova et al., 2022), are guaranteed to achieve a low simple regret, they are not necessarily guaranteed to do so with low evaluation costs. This is important because—as our experiments will show—UCB–LCB will often incur high evaluation costs, resulting in a high cost-adjusted simple regret.

Our work builds upon the Bayesian-optimal Pandora's Box decision principle underlying the PBGI acquisition function and extends it to the adaptive stopping setting in correlated Bayesian optimization. Furthermore, we show that an existing stopping rule proposed for the expected improvement (EI) acquisition function in the classical (non–cost-aware) setting (Nguyen et al., 2017) is equivalent to the PBGI stopping rule, which provides a principled extension of that rule to the cost-aware setting. Our stopping rule can therefore naturally be paired with both PBGI and (Log)EI(PC), but it does not automatically extend to acquisition functions based on other design principles (e.g., KG, MES), which would require their own matched stopping rules. Our specific contributions are as follows:

1. *A Novel Cost-Aware Stopping Rule.* We present an adaptive stopping rule derived from Pandora's box theory, which also naturally extends to the EI design principle, establishing a unified and principled framework applicable to classic and cost-aware Bayesian optimization.

2. *Theoretical Guarantees.* We prove in Theorem 2 that our stopping rule, when paired with the PBGI or LogEIPC acquisition function, satisfies a theoretical upper bound on the expected cost-adjusted simple regret, which constitutes the first theoretical guarantee of this type for any adaptive stopping rule for Bayesian optimization.

3. *Empirical Validation.* We conduct a systematic empirical evaluation across multiple acquisition-function—stopping-rule pairs in cost-aware Bayesian optimization. Our results show that pairing our proposed stopping rule with PBGI or LogEIPC usually matches or outperforms other pairs of acquisition functions and stopping rules.

## 2 BAYESIAN OPTIMIZATION AND ADAPTIVE STOPPING

In black-box optimization, the goal is to find an approximate optimum of an unknown objective function $f : X \to \mathbb{R}$ using a limited number of function evaluations at points $x_1, \ldots, x_T \in X$ where $X$ is the search space and $T$ is a given search budget, potentially chosen adaptively. The convention measures performance in terms of *expected simple regret* (Garnett, 2023, Sec. 10.1), given by

$$\mathcal{R} = \mathbb{E}\left[\min_{1 \leq t \leq T} f(x_t) - \inf_{x \in X} f(x)\right] \tag{1}$$

where the expectation is taken over all sources of randomness, including the sequence of points $x_1, \ldots, x_T$ selected by the algorithm. Bayesian optimization approaches this problem by building a probabilistic model of $f$—typically a Gaussian process (GP) (Rasmussen & Williams, 2006), conditioned on the observed data points $(x_t, y_t)_{t=1}^T$, where $y_t = f(x_t)$. For each iteration $t = 1, \ldots, T$, an acquisition function $\alpha_t : X \to \mathbb{R}$ then guides the selection of new samples by carefully balancing the exploration-exploitation tradeoff arising from uncertainty about $f$.

### 2.1 COST-AWARE BAYESIAN OPTIMIZATION

*Cost-aware Bayesian optimization* (Lee et al., 2020; Astudillo et al., 2021) extends the above setup to account for the fact that evaluation costs can vary across the search space. For instance, in the hyperparameter tuning example of Section 1, costs can vary according to the time needed to train a

machine learning model under a given set of hyperparameters $x$. Cost-aware Bayesian optimization handles this by introducing a cost function $c : X \to \mathbb{R}^+$, which may be known or unknown ahead. The cost function, or observed costs, are then used to construct the acquisition function $\alpha_t$.

In this work, we focus primarily on the *cost-per-sample* formulation (Chick & Frazier, 2012; Cashore et al., 2016; Xie et al., 2024) of cost-aware Bayesian optimization, which seeks methods with stopping time $\tau$, not necessarily fixed, that achieve a low *expected cost-adjusted simple regret*

$$\mathcal{R}_c = \mathbb{E}\left[\underbrace{\min_{1 \leq t \leq \tau} f(x_t) - \inf_{x \in X} f(x)}_{\text{simple regret}} + \underbrace{\sum_{t=1}^{\tau} c(x_t)}_{\text{cumulative cost}}\right]. \tag{2}$$

One can also work with the *expected budget-constrained* formulation (Xie et al., 2024), which incorporates budget constraints explicitly, and seeks algorithms which achieve a low expected simple regret under an expected evaluation budget. Here, performance is evaluated in terms of

$$\mathcal{R} = \mathbb{E}\left[\min_{1 \leq t \leq \tau} f(x_t) - \inf_{x \in X} f(x)\right] \qquad \text{where } x_1, \ldots, x_\tau \text{ satisfy} \qquad \mathbb{E}\sum_{t=1}^{\tau} c(x_t) \leq B. \tag{3}$$

The stopping rules we study can be applied in this setting as well: we discuss this in Section 3.1.

For both settings, we work with a few acquisition functions—chiefly, *log expected improvement per cost (LogEIPC)* (Ament et al., 2023) and the *Pandora's Box Gittins Index (PBGI)* (Xie et al., 2024). Given the posterior distribution of $f$ at iteration $t$, LogEIPC is defined as

$$\alpha_t^{\text{LogEIPC}}(x) := \log \frac{\text{EI}_{f|x_{1:t}, y_{1:t}}(x; y_{1:t}^*)}{c(x)}, \tag{4}$$

i.e., the logarithm of the expected improvement divided by the cost; and PBGI is defined as

$$\alpha_t^{\text{PBGI}}(x) := \begin{cases} g & \text{where } g \text{ solves} \quad \text{EI}_{f|x_{1:t}, y_{1:t}}(x; g) = c(x), & x \notin \{x_1, \ldots, x_t\} \\ f(x), & x \in \{x_1, \ldots, x_t\} \end{cases} \tag{5}$$

where the value $g$ is the threshold at which the expected improvement equals the cost[1], $y_{1:t}^* = \min_{1 \leq s \leq t} y_s$ is the best value observed in the first $t$ iterations, and $\text{EI}_\psi(x; y) = \mathbb{E}\left[\max(y - \psi(x), 0)\right]$ is the expected improvement at point $x$ with respect to some random function $\psi : X \to \mathbb{R}$ and a baseline value $y$. We also consider classical cost-unaware acquisition functions—*lower confidence bound (LCB)* and *Thompson sampling (TS)*; see Garnett (2023) for details.

## 2.2 Adaptive Stopping Rules

To the best of our knowledge, stopping rules for Bayesian optimization typically do not incorporate cost explicitly into the stopping criterion. We broadly categorize existing methods as follows:

*Criteria based on convergence or significance of improvement.* This includes empirical convergence, namely stopping when the best value remains unchanged for a fixed number of iterations, or the *global stopping strategy (GSS)* (Bakshy et al., 2018), which stops when the improvement is no longer statistically significant relative to the inter-quartile range (i.e., the range between the 25th percentile and the 75th percentile) of prior observations. In the multi-fidelity setting, Foumani & Bostanabad (2025) proposed stopping when the high-fidelity surrogate's estimated optimum has stabilized over a window of iterations, which is related to cost-awareness but differs from our setting with explicitly varying evaluation costs.

*Acquisition-based criteria.* This includes stopping rules built from acquisition functions such as the *probability of improvement (PI)* (Lorenz et al., 2015), *expected improvement (EI)* (Locatelli, 1997; Nguyen et al., 2017; Ishibashi et al., 2023), and *knowledge gradient (KG)* (Frazier & Powell, 2008). These approaches typically stop when the acquisition value falls below a fixed threshold—a predetermined constant, median of the initial acquisition values, or the cost of sampling—but typically assume uniform costs and do not adapt to evaluation costs which can vary across the search space.

---

[1]For an evaluated point $x$, we set $g = f(x)$ since the posterior at $x$ collapses to a point mass at the observed value $f(x)$ and its evaluation cost can be treated as 0.

*Regret-based criteria.* This includes stopping rules based on confidence bounds such as *UCB-LCB* (Makarova et al., 2022) and the *gap of expected minimum simple regrets* (Ishibashi et al., 2023). These stop when certain estimated regret bounds fall below a preset or data-driven threshold. The related *probabilistic regret bound (PRB)* stopping rule (Wilson, 2024) stops when estimated simple regret is below a small threshold $\epsilon$ with confidence $1 - \delta$.

In settings beyond Bayesian optimization, Chick & Frazier (2012) have proposed a cost-aware stopping rules for finite-domain sequential sampling problems with independent values. Although this formulation allows for varying costs, it does not extend to general Bayesian optimization settings which use correlated GP models.

## 3 A Stopping Rule Based on the Pandora's Box Gittins Index

In this work, we propose a new stopping rule tailored for two state-of-the-art acquisition functions used in cost-aware Bayesian optimization: LogEIPC and PBGI, introduced in Section 2. As discussed by Xie et al. (2024), these acquisition functions are closely connected, arising from two different approximations of the intractable dynamic program which defines the Bayesian-optimal policy for cost-aware Bayesian optimization. We now show that this connection can be used to obtain a principled stopping criterion to be paired with both acquisition functions.

**A stopping rule for PBGI.** To derive a stopping rule for PBGI, consider first the Pandora's Box problem, from which it is derived. Pandora's Box can be seen as a special case of cost-per-sample Bayesian optimization, as introduced in Section 2, where $X = \{1, \ldots, N\}$ is a discrete space and $f$ is assumed uncorrelated. In this setting, the observed values do not affect the posterior distribution—meaning, we have $f(x') \mid x_{1:t}, y_{1:t} = f(x')$ at all unobserved points $x'$—and thus the acquisition function value at point $x$ is time-invariant and can be written simply as $\alpha^{\mathrm{PBGI}}(x)$, where $\mathrm{EI}_f(x; \alpha^{\mathrm{PBGI}}(x)) = c(x)$. Following Weitzman (1979), one can show using a Gittins index argument that selecting $x_t$ to minimize $\alpha^{\mathrm{PBGI}}$ is Bayesian-optimal under minimization—the algorithm which does so achieves the smallest expected cost-adjusted simple regret, among all adaptive algorithms.

One critical detail applied in the optimality argument of Weitzman (1979) is that the policy defined by $\alpha^{\mathrm{PBGI}}$ is *not* Bayesian-optimal for any fixed deterministic $T$. Instead, it is optimal only when the stopping time $T$ is chosen according to the condition

$$\min_{x \in X \setminus \{x_1, \ldots, x_t\}} \alpha^{\mathrm{PBGI}}(x) \geq y_{1:t}^*, \tag{6}$$

where $\geq$ can be replaced by $>$[2], and as before, $y_{1:t}^*$ is the best value observed so far.

In the correlated setting we study, Xie et al. (2024) extend $\alpha^{\mathrm{PBGI}}$ to define an acquisition function by recomputing it at each time step $t$ based on the posterior mean and variance, which defines $\alpha_t^{\mathrm{PBGI}}$ as in Equation (5). In order to also extend the associated stopping rule, a subtle design choice arises: should we use $\alpha_{t-1}^{\mathrm{PBGI}}$ (before posterior update) or $\alpha_t^{\mathrm{PBGI}}$ (after posterior update) in Equation (6)? While prior theoretical work (Gergatsouli & Tzamos, 2023) adopts the former choice, we instead propose the latter, for two main reasons.

First, we argue it more faithfully reflects the intuition behind Weitzman's original stopping rule. In the independent setting, $\alpha^{\mathrm{PBGI}}(x)$ represents a kind of *fair value* for point $x$ (Kleinberg et al., 2016). For an evaluated point $x \in \{x_1, \ldots, x_t\}$, this is simply the observed function value. For an unevaluated point $x \notin \{x_1, \ldots, x_t\}$, the fair value reflects uncertainty in $f(x)$ and the cost $c(x)$. The stopping rule Equation (6) then says to *stop when the best fair value is among the already-evaluated points*—namely, when no point provides positive expected gain relative to its cost if evaluated in the next round, conditioned on current observations. In the correlated setting, the fair value naturally extends to $\alpha_t^{\mathrm{PBGI}}$, because this incorporates all known information at a given time. Second, we show in Appendix D.2 that using $\alpha_t^{\mathrm{PBGI}}$ yields tangible empirical gains in cost-adjusted regret.

**Connection with LogEIPC.** The above reasoning may at first seem to be fundamentally tied to the Pandora's Box problem and its Gittins-index-theoretic properties. We now show that it admits a

---

[2]Following Xie et al. (2024, Appendix B), optimality holds under any tie-breaking rule in the cost-per-sample setting, but only under a carefully-chosen stochastic tie-breaking rule in the expected budget-constrained setting. For simplicity, we use the stopping-as-early-as-possible tie-breaking rule throughout this paper.

second interpretation in terms of log expected improvement per cost, which arises from a completely different one-time-step approximation to the Bayesian-optimal dynamic program for cost-aware Bayesian optimization in the general correlated setting.

To show this, we start with definitions above and apply a sequence of transformations. Recall that for any unevaluated point $x \notin \{x_1, \ldots, x_t\}$, $\alpha_t^{\text{PBGI}}(x)$ is defined in Equation (5) to be the solution to

$$\text{EI}_{f|x_{1:t},y_{1:t}}(x; \alpha_t^{\text{PBGI}}(x)) = c(x) \tag{7}$$

and since $\text{EI}_\psi(x; y)$ is strictly increasing in $y$, any inequality involving $\alpha_t^{\text{PBGI}}(x; c)$ can be lifted through the EI function without changing its direction, which means

$$\alpha_t^{\text{PBGI}}(x) \geq y_{1:t}^* \qquad \text{holds if and only if} \qquad \text{EI}_{f|x_{1:t},y_{1:t}}(x; y_{1:t}^*) \leq c(x). \tag{8}$$

This implies that the PBGI stopping rule from Equation (6) is equivalent to stopping when

$$\text{EI}_{f|x_{1:t},y_{1:t}}(x; y_{1:t}^*) \leq c(x) \quad \text{for all } x \in X \backslash \{x_1, \ldots, x_t\}. \tag{9}$$

meaning *stop when no point's expected improvement is worth its evaluation cost*. Rearranging the inequality and taking logs, we can rewrite this condition using the LogEIPC acquisition function [3]:

$$\max_{x \in X \backslash \{x_1, \ldots, x_t\}} \alpha_t^{\text{LogEIPC}}(x; y_{1:t}^*) \leq 0. \tag{10}$$

We call the stopping rule given by the equivalent conditions (Equations (6), (9) and (10)) the *PBGI/LogEIPC stopping rule*. Figure 1 gives an illustration of how the rule behaves in a simple setting, demonstrating that it is more conservative—preferring to stop earlier—when the cost is high.

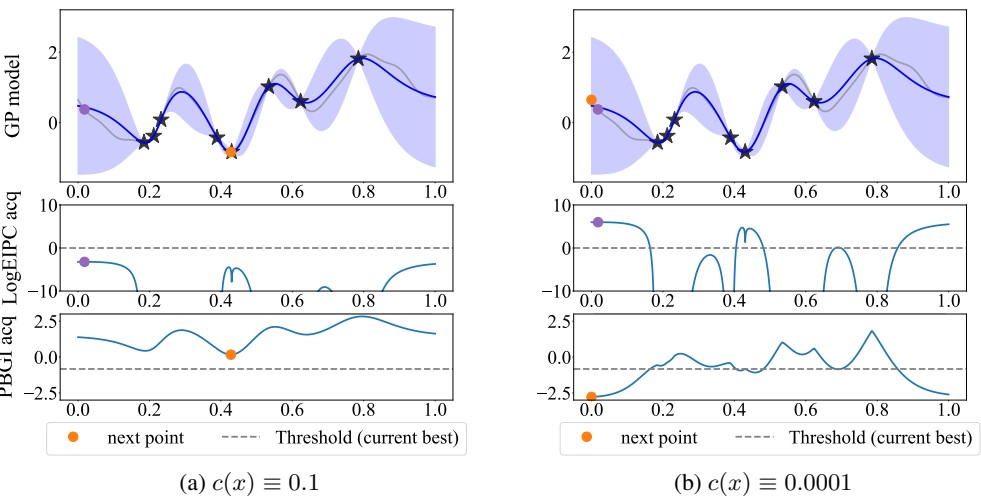

(a) $c(x) \equiv 0.1$        (b) $c(x) \equiv 0.0001$

Figure 1: Illustration of the PBGI/LogEIPC stopping rule under a uniform-cost setting. When the cost-per-sample is large ($c(x) \equiv 0.1$), the maximum LogEIPC acquisition value falls below the threshold 0.0 and the minimum PBGI acquisition value exceeds the current best observed value, indicating stopping; when the cost-per-sample is small ($c(x) \equiv 0.0001$), the maximum LogEIPC acquisition value remains above the threshold 0.0 and the minimum PBGI acquisition value is smaller than the current best observed value, indicating no stopping.

### 3.1 THEORETICAL GUARANTEES ON COST-ADJUSTED SIMPLE REGRET

The connection between PBGI and LogEIPC in fact goes beyond a shared stopping rule. In Lemma 1, we prove that when paired with this stopping rule, both acquisition functions guarantee that at each iteration before stopping, the expected improvement at the selected point is at least its evaluation cost.

---

[3]Excluding evaluated points is not theoretically required but often beneficial in practice, as numerical stability adjustments can cause their expected improvement to remain positive.

**Lemma 1.** *Let $X$ be compact, and let $f : X \to \mathbb{R}$ be a random function with prior mean $\mu(\cdot)$. Consider a Bayesian optimization algorithm that begins at some initial point $x_1 \in X$ with cost $C = c(x_1)$, acquires subsequent points using either the PBGI or LogEIPC acquisition function, and terminates according to the PBGI/LogEIPC stopping rule. Let $\tau = \min_{t \geq 1}\{\sup_{x \in X} \alpha_t^{\mathrm{LogEIPC}}(x) < 0\}$ be the algorithm's stopping time, and denote the posterior expected improvement function by $\alpha_t^{\mathrm{EI}}(x) = \mathrm{EI}_{f|x_{1:t}, y_{1:t}}(x; y_{1:t}^*)$. Then, for all $t < \tau$, $\alpha_t^{\mathrm{EI}}(x_{t+1}) \geq c(x_{t+1})$.*

As a consequence of this claim, we show in Theorem 2 that the PBGI/LogEIPC stopping rule, when paired with PBGI or LogEIPC, achieves cost-adjusted simple regret no worse than stopping immediately after the initial evaluation. Although this may sound trivial, many acquisition function–stopping rule pairs fail to satisfy this *no-worse-than-immediate* property in practice (see Figures 2 and 3), largely because most stopping rules are designed with simple regret alone in mind and do not explicitly account for evaluation costs. To our knowledge, this is the first formal guarantee on cost-adjusted simple regret for Bayesian optimization, generalizing beyond simpler settings such as Pandora's Box, which assumes independent and discrete evaluations. All proofs in this section are in Appendix B.

**Theorem 2.** *Consider the setting and algorithm specified in Lemma 1. Let $U := \mu(x_1) - \mathbb{E}[\min_{x \in X} f(x)] < \infty$, then the algorithm's expected cost-adjusted simple regret is bounded by*

$$\mathbb{E}\left[ y_{1:\tau}^* - \min_{x \in X} f(x) + \sum_{t=1}^{\tau} c(x_t) \right] \leq \mathbb{E}\big[ \underbrace{y_1 - \min_{x \in X} f(x) + c(x_1)}_{\text{cost-adjusted regret of immediate stopping}} \big] = U + C. \tag{11}$$

This result yields a bound on expected cumulative cost and, under the natural assumption that practical evaluations have a positive minimum cost, a high-probability finite-time termination guarantee.

**Corollary 3.** *Consider the setting and algorithm specified in Lemma 1. Then the expected cumulative cost of the algorithm is bounded by $\mathbb{E}\left[\sum_{t=1}^{\tau} c(x_t)\right] \leq U + C$. Further, if evaluation costs are uniformly bounded below by a constant $c_0 > 0$, i.e., $c(x) \geq c_0, \forall x \in X$, then for any $\delta \in (0, 1)$, the algorithm terminates in at most $\frac{U+C}{\delta \cdot c_0}$ iterations with probability $1 - \delta$.*

### 3.1.1 Implication in the Budget-constrained Setting

We first note that in the discrete Pandora's Box setting, under an expected budget constraint $B$, minimizing $\alpha_t^{\mathrm{PBGI}}(x)$ is Bayesian-optimal: it achieves the lowest *expected simple regret, among all algorithms* satisfying $\mathbb{E}\left[\sum_{t=1}^{\tau} c(x_t)\right] \leq B$, and the cost function used in defining $\alpha_t^{\mathrm{PBGI}}(x)$ is $\lambda c(x)$ for some cost-scaling factor $\lambda$ which depends on $B$ (Xie et al., 2024, Theorem 2).

This result, at first, appear to be completely Pandora's-Box-theoretic: it requires $X$ to be discrete and $f$ to be independent. In the more general correlated setting of cost-aware Bayesian optimization, however, Bayesian optimality of PBGI may no longer hold, and the relationship between $B$ and the choice of $\lambda$ is not immediately clear. Theorem 2 helps bridge this gap: it provides an upper bound on the expected cumulative cost up to the stopping time, which in turn yields the following principled choice of $\lambda$ that ensures compliance with the budget constraint.

**Corollary 4.** *Consider the setting, algorithm, and notation specified in Lemma 1 and Theorem 2, but with costs rescaled by a factor $\lambda > 0$: both the acquisition values and the stopping conditions are computed using $\lambda c(\cdot)$. If the cost-scaling factor is set to $\lambda = \frac{U}{B-C}$, then the algorithm's expected cumulative unscaled cost satisfies $\mathbb{E}[\sum_{t=1}^{\tau} c(x_t)] \leq B$.*

If this choice of $\lambda$ proves overly conservative—leading to underspending within the budget—it can be paired with the PBGI-D variant of Xie et al. (2024), which starts with $\lambda = \lambda_0$ and halves it each time stopping is triggered. Choosing $\lambda_0 = U/(B - C)$ aligns the initial fixed-$\lambda$ phase with the budget in expectation, while ensuring that the adaptive decay is activated as designed. In Figure 10 in Appendix D, we show PBGI-D with this choice of $\lambda_0$ is competitive.

## 3.2 Practical Implementation Considerations for PBGI/LogEIPC Stopping Rule

**Expressing objective and cost in common units.** Evaluation costs are often measured in different units than the objective, such as time versus accuracy. To compare them directly, we rescale costs by a constant $\lambda > 0$, the conversion rate between cost and objective improvement. For instance, if

one is willing to spend 1000 seconds to reduce test error by 0.01, then $\lambda = 10^{-5}$. Importantly, in the cost-per-sample setting we mainly study, $\lambda$ is set by the problem provider rather than tuned by the user, whereas the budgeted setting without a natural unit conversion is discussed in Section 3.1.1.

**Unknown costs.** In practice, evaluation costs are often not known in advance. They can be modeled either (i) deterministically, using domain knowledge (e.g., proportional to model size in hyperparameter tuning), or (ii) stochastically, via a GP over log costs. In Section 4, we present empirical benchmark results under both modeling approaches. For the stochastic approach, we follow Astudillo et al. (2021, Proposition 2) to model $\ln c(x)$ as a GP with posterior mean $\mu_{\ln c}$ and variance $\sigma_{\ln c}^2$. To compute LogEIPC or PBGI, as well as their related stopping rules (including prior acquisition-based ones and ours), we replace $c(x)$ in Equations (4), (5) and (9) with $\mathbb{E}[c(x)] = \exp(\mu_{\ln c}(x) + (\sigma_{\ln c}(x))^2/2)$. See Appendix D.3 for further discussion.

**Preventing spurious stops.** Although our stopping rule has theoretical guarantees, in practice, it—like other adaptive stopping rules—can still suffer from spurious stops caused by two main sources. First, early in the optimization process, the GP model parameters often fluctuate significantly as new data points are collected, causing unstable stopping signals. To mitigate this, we enforce a stabilization period consisting of the first several evaluations, during which we do not allow any stopping rule to trigger. Second, imperfect optimization of the acquisition function—which is especially common in higher-dimensional search spaces—can lead to misleading stopping signals. To handle this, we use a *debounce* strategy, requiring the stopping rule to consistently indicate stopping over several consecutive iterations before stopping optimization. See Figure 4 for an illustration.

## 4 EXPERIMENTS

To evaluate our proposed PBGI/LogEIPC stopping rule, we design three complementary sets of experiments that progressively test its performance. First, we consider an idealized, low-dimensional Bayesian regret setting in which the GP model exactly matches the true objective. This controlled environment allows us to isolate the effect of different stopping rules without interference from modeling or optimization errors. Then, we move to a higher-dimensional Bayesian regret setting where each acquisition-function minimization must be approximated via a numerical optimizer. Finally, we test in practical scenarios with potential objective model mismatch, using the LCBench hyperparameter tuning benchmark and the NATS neural architecture size search benchmark. In each case, we compare pairs of acquisition functions and stopping rules, which we describe next.

**Acquisition functions.** We consider four common acquisition functions that were discussed in Section 2: LogEIPC, PBGI, LCB, and TS—chosen for their competitive performance, computational efficiency, and close connections to existing stopping rules.

**Baselines.** We compare the proposed PBGI/LogEIPC stopping rule against several stopping rules from prior work: *UCB–LCB* (Makarova et al., 2022), *LogEIPC-med* (Ishibashi et al., 2023), *SRGap-med* (Ishibashi et al., 2023), and *PRB* (Wilson, 2024). We also include two simple heuristics used in practice: *Convergence* and *GSS*. UCB–LCB stops once the gap between upper and lower confidence bounds falls below a configurable threshold $\theta$. LogEIPC-med stops when the log expected improvement per cost drops beneath $\log(\eta)$ plus the median of its initial $I$ values. SRGap-med stops when the gap of the expected minimum simple regret falls below $\chi$ times the median of its initial $I$ values. PRB triggers once a probabilistic regret bound satisfies the regret tolerance $\epsilon$ and confidence $\delta$ parameters. Convergence stops as soon as the best observed value remains unchanged for $w$ iterations. GSS stops if the recent improvement is less than $\phi \times \text{IQR}$ over the past $w$ trials where IQR denotes the inter-quartile range of current observations. Finally, we include two reference baselines: *Immediate*, which stops right after the initial evaluation (see Section 3.1), and *Hindsight*, which, for each trial, selects the stopping time that yields the lowest cost-adjusted simple regret in hindsight, thus providing a lower bound on achievable performance.

In our experiments, unless specified otherwise, we follow parameter values recommended in the literature, and set $\theta = 0.01$, $\eta = 0.01$, $\chi = 0.01$, $I = 20$, $\epsilon = 0.1$ for Bayesian regret experiments, $\epsilon = 0.5\%$ of the best test error for empirical experiments, $\delta = 0.05$, $w = 5$, and $\phi = 0.01$. Each experiment is repeated with 50 random seeds to assess variability, and we report the mean with error

bars (2 times the standard error) for each stopping rule. Each trial, in the sense of a run with a distinct random seed, is capped at a fixed number of iterations; if a stopping rule is not triggered within this limit, the stopping time is set to the cap. Details are provided in Appendix C.

## 4.1 BAYESIAN REGRET

We first evaluate our PBGI/LogEIPC stopping rule on random functions sampled from prior. Specifically, objective functions are sampled from Gaussian processes with Matérn 5/2 kernels with length scale 0.1, defined over spaces of dimension $d = 1$ and $d = 8$. We consider a variety of evaluation cost function types, including uniform costs, linearly increasing costs in terms of parameter magnitude, and periodic costs that vary non-monotonically over the domain.

In the 1-dimensional experiments, we perform an exhaustive grid search over $[0, 1]$ to isolate the effect of stopping behavior from numerical optimization. In the 8-dimensional setting, due to instability from higher dimensionality, we apply moving average with a window size of 20 when computing the PBGI/LogEIPC stopping condition. See Figure 4 in Appendix C for an illustration. More details on our experiment setup, and computational considerations are provided in Appendices C and D.

Figure 2 shows a comparison of cost-adjusted regret for acquisition function and stopping rule pairings under different cost-scaling factors $\lambda$ in the 1-dimensional setting and under different cost regimes (uniform, linear, periodic) in 8-dimensional setting.

In the 1-dimensional setting, the Bayesian optimization problem is relatively straightforward and all acquisition functions attain nearly identical hindsight optima, independent of cost scale or cost type. From the plot, our PBGI/LogEIPC stopping rule pairing with LogEIPC or PBGI acquisition function delivers competitive performance for each $\lambda$, and is particularly strong in handling high-cost scenarios—those with large $\lambda$.

In the 8-dimensional experiments, however, clear gaps emerge: Under uniform and linear costs, LogEIPC, PBGI, and LCB perform similarly and substantially outperform TS, while in the periodic case, LogEIPC and PBGI outperform LCB and TS. In every cost regime, combining our stopping rule with the LogEIPC, PBGI, or LCB acquisition function yields cost-adjusted regret nearly matching the hindsight optimal. This shows our PBGI/LogEIPC stopping rule to be relatively robust against challenges in acquisition function optimization. Further discussions of our experiments across all cost types and cost-scaling factors can be found in Appendix D.

## 4.2 EMPIRICAL AUTOMATED MACHINE LEARNING BENCHMARKS

We then evaluate on two empirical benchmarks based on use cases from hyperparameter optimization and neural architecture search: LCBench (Zimmer et al., 2021) and NATS-Bench (Dong et al., 2021).

LCBench provides training data of 2,000 hyperparameter configurations evaluated on 35 OpenML Vanschoren et al. (2014) datasets. In the *unknown-cost* experiments, the cost is the full (200-epoch) training time. For the known-cost setting, we observe that training time to scale approximately linearly with the number of model parameters. Based on a linear regression fit (see Appendix C), we adopt $\alpha p$ as a proxy of training time, where $p$ is the number of model parameters and $\alpha$ is a dataset-specific coefficient estimated from the regression.

NATS-Bench provides a search space of 32,768 neural architectures varying in channel sizes across layers, evaluated on three datasets. As in LCBench, for the known-cost experiments, we approximate the full runtime (training + validation time) at 90 epochs using $\alpha F + \beta$, where $F$ is the number of floating point operations; see Appendix C for details.

We report the mean and error bars (2 times standard error) of the *cost-adjusted simple regret*, where we consider the evaluation costs to be some representative cost-scaling factor $\lambda$ (see Section 3.2) multiplied with cumulative runtime. Simple regret is computed as the difference between (a) test error of the configuration with best validation error at the stopping time and (b) the best test error over all configurations. We present the results using proxy runtime here, and defer to Appendix D the additional results under (i) the *cost model mismatch* scenario (proxy runtime used during Bayesian optimization but actual runtime used for final performance evaluation), and (ii) the *unknown-cost* scenario (actual runtime used during Bayesian optimization).

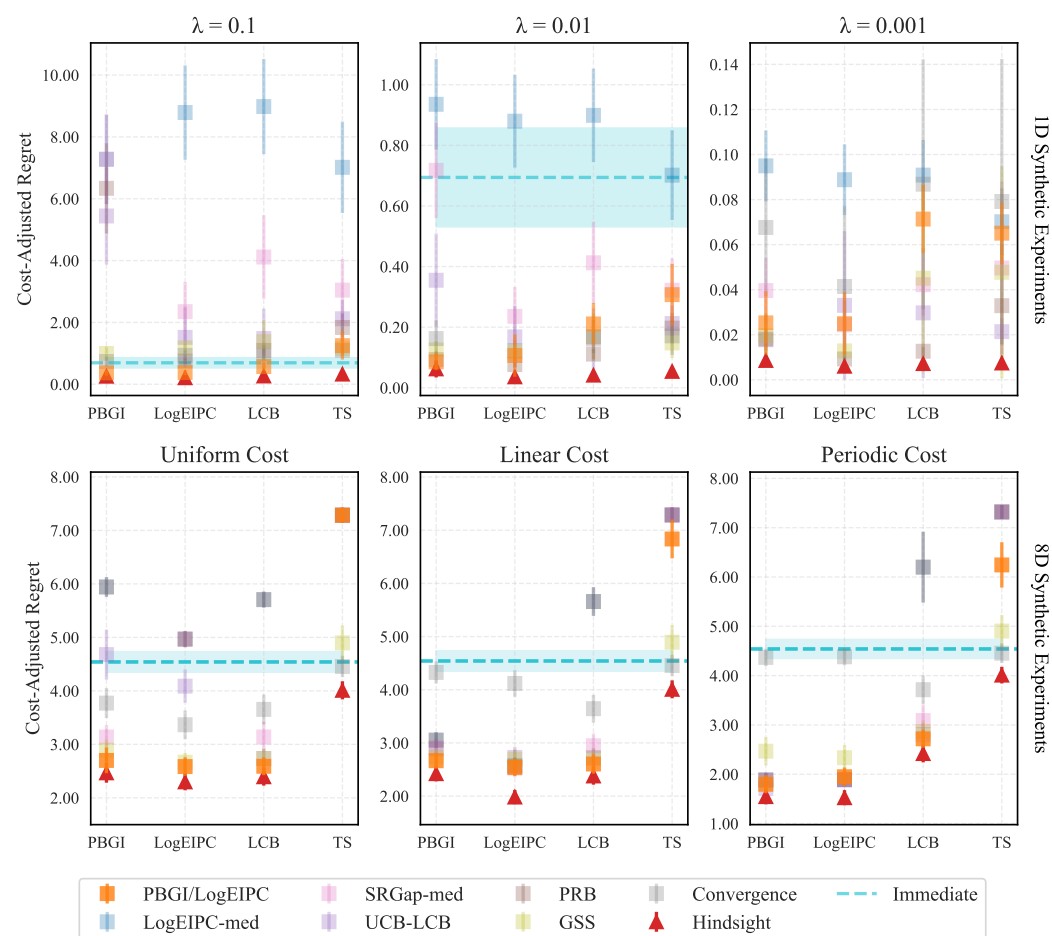

Figure 2: Cost-adjusted simple regret across acquisition-stopping rule pairs in 1D and 8D Bayesian regret setting. In 1D, objective functions are sampled from a GP with a Matérn-5/2 kernel and a linear cost function scaled by $\lambda = 0.1, 0.01, 0.001$. The Immediate baseline is omitted at $\lambda = 0.001$ due to its much higher regret (mean 0.6942, error bar [0.5314, 0.8570]). In 8D, objective functions are also drawn from a GP with a Matérn-5/2 kernel, using three cost functions scaled by $\lambda = 0.01$.

Figure 3 presents performance of acquisition function–stopping rule pairs on LCBench and NATS-Bench. For LCBench, we report min–max normalized cost-adjusted simple regret (see definition in Appendix D) aggregated across 35 datasets in Figure 3 evaluated under three representative[4] cost-scaling values. For NATS-Bench, we report cost-adjusted simple regret under $\lambda = 10^{-5}$, with additional results under two other cost-scaling factors shown in Figures 14 to 16 in Appendix D.

Per-dataset LCBench results are provided in Figures 19 to 21 in the appendix. Across these 35 tasks, around 75% of them show our PBGI/LogEIPC stopping rule performing competitively when paired with either PBGI or LogEIPC. The remaining outliers are mostly (aside from two exceptions) very small datasets with <10000 instances that might lead to severe model misspecification.

Additional results under the cost model mismatch setting and the unknown-cost setting are provided in Figures 17 and 18 in Appendix D, along with comparisons between our adaptive stopping rule and fixed-iteration baselines. Overall, our PBGI/LogEIPC stopping rule consistently performs strongly in terms of cost-adjusted simple regret, when paired with either PBGI or LogEIPC. We also report, in Table 1 of Appendix D, how often each stopping rule fails to trigger within the 200-iteration cap: baselines such as SRGap-med and UCB–LCB often fail to stop early—particularly on NATS-Bench—whereas our stopping rule reliably stops before the cap.

---

[4]These $\lambda$ values are chosen to avoid degenerate cases—neither so large that the policy stops after only a few evaluations (e.g., $\lambda = 10^{-4}$ on NATS-Bench, see Figure 17) nor so small that evaluations are effectively free.

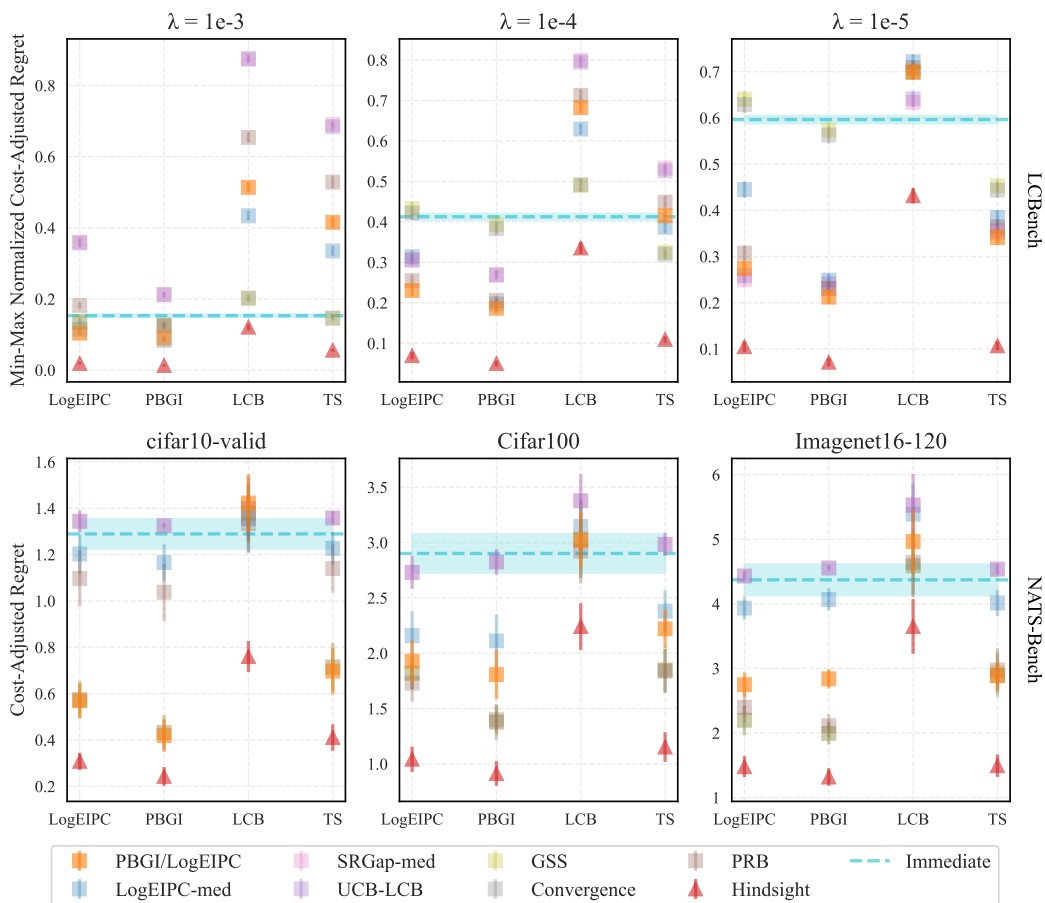

Figure 3: Cost-adjusted simple regret across acquisition function–stopping rule pairs on LCBench and NATS-Bench. The objective is to minimize validation error on classification tasks, with scaled proxy runtime as evaluation cost, scaled by representative values of $\lambda$ ($10^{-3}, 10^{-4}, 10^{-5}$ for LCBench and $10^{-5}$ for NATS-Bench). For LCBench, results are aggregated across 35 datasets using min–max normalization. Our PBGI/LogEIPC stopping rule, when paired with either LogEIPC or PBGI, typically ranks among the top 3 pairs and closely approaches the hindsight optimal on LCBench and on cifar10-valid in NATS-Bench, but slightly worse on the other two NATS datasets.

As we transition from model match to model mismatch, where the true objective function does not align perfectly with the GP prior, we find that our stopping rule, when paired with the PBGI acquisition function, continues to deliver performance close to the hindsight optimal, except on two NATS datasets likely affected by severe mismatch. In contrast, pairing with the LogEIPC acquisition function is less competitive. This degradation appears to stem from the relative advantage of PBGI over LogEIPC in higher dimensions or misspecified settings: as shown in Figure 10 in Appendix D, PBGI maintains stronger performance under misspecification, thus contributing to the improved overall cost-adjusted regret. Thus, while our stopping rule is broadly robust, the choice of acquisition function remains an important consideration when facing objective model mismatch.

## 5 CONCLUSION

We develop the *PBGI/LogEIPC stopping rule* for Bayesian optimization with varying evaluation costs. Paired with either the PBGI or LogEIPC acquisition function, it (a) satisfies a theoretical guarantee bounding the expected cost-adjusted simple regret (Section 3.1), and (b) shows strong empirical performance in terms of cost-adjusted simple regret (Section 4). We believe our framework can be extended to settings involving noisy, multi-fidelity or batched evaluations, as well as alternative objective formulations—for instance, applying a sigmoid transformation to test error rather than a linear one, to reflect real-world user preferences that shift sharply once error falls below a threshold.

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

## A  LLM USAGE DISCLOSURE

We used large language models (LLMs) to assist with writing and editing this paper (e.g., improving clarity and readability of drafts). LLMs were also used to help polish proofs, to facilitate experiment scripting, and to generate plotting code. All technical content, research ideas, and final implementations were developed, verified, and approved by the authors.

## B  THEORETICAL ANALYSIS AND CALCULATIONS

In the following lemma, we prove a point-wise lower bound on the expected improvement before stopping for our recommended pairing of PBGI/LogEIPC stopping rule paired with either PBGI or LogEIPC acquisition function.[5]

**Lemma 1.** *Let $X$ be compact, and let $f : X \to \mathbb{R}$ be a random function with prior mean $\mu(\cdot)$. Consider a Bayesian optimization algorithm that begins at some initial point $x_1 \in X$ with cost $C = c(x_1)$, acquires subsequent points using either the PBGI or LogEIPC acquisition function, and terminates according to the PBGI/LogEIPC stopping rule. Let $\tau = \min_{t \geq 1} \{ \sup_{x \in X} \alpha_t^{\mathrm{LogEIPC}}(x) < 0 \}$ be the algorithm's stopping time, and denote the posterior expected improvement function by $\alpha_t^{\mathrm{EI}}(x) = \mathrm{EI}_{f|x_{1:t}, y_{1:t}} (x; y_{1:t}^*)$. Then, for all $t < \tau$, $\alpha_t^{\mathrm{EI}}(x_{t+1}) \geq c(x_{t+1})$.*

*Proof.* While stopping has not occurred, meaning $t < \tau$, by the stopping criteria definition we have $\max_{x \in X} \alpha_t^{\mathrm{EI}}(x)/c(x) \geq 1$. Hence, there exists at least one point with $\alpha_t^{\mathrm{EI}}(x)/c(x) \geq 1$. We now argue for each algorithm.

*PBGI.* For each $x$ we defined a threshold $\alpha_t^{\mathrm{PBGI}}(x)$ by $\mathrm{EI}_{f|x_{1:t}, y_{1:t}} (x; \alpha_t^{\mathrm{PBGI}}(x)) = c(x)$. Since $\mathrm{EI}_{f|x_{1:t}, y_{1:t}}$ is increasing in its second argument,

$$y_{1:t}^* \geq \alpha_t^{\mathrm{PBGI}}(x) \iff \alpha_t^{\mathrm{EI}}(x)/c(x) \geq 1. \tag{12}$$

The existence of a point with ratio at least 1 therefore implies that the set $\mathcal{S}_t = \{ x : y_{1:t}^* \geq \alpha_t^{\mathrm{PBGI}}(x) \}$ is non-empty. PBGI chooses $x_{t+1}$ with the *smallest* threshold, and thus

$$\alpha_t^{\mathrm{EI}}(x_{t+1}) = \mathrm{EI}_{f|x_{1:t}, y_{1:t}} (x_{t+1}; y_{1:t}^*) \geq \mathrm{EI}_{f|x_{1:t}, y_{1:t}} (x_{t+1}; \alpha_t^{\mathrm{PBGI}}(x_{t+1})) = c(x_{t+1}). \tag{13}$$

*(Log)EIPC.* By definition $x_{t+1}$ maximizes $\log(\alpha_t^{\mathrm{EI}}(x)/c(x))$ and $\alpha_t^{\mathrm{EI}}(x)/c(x)$, hence $\alpha_t^{\mathrm{EI}}(x_{t+1}) \geq c(x_{t+1})$.

Thus for *both* algorithms, we have

$$\alpha_t^{\mathrm{EI}}(x_{t+1}) \geq c(x_{t+1}), \qquad \text{for all } t < \tau. \tag{14}$$

$\square$

We can now use Lemma 1 to prove the following theorem, where we show that our PBGI/LogEIPC stopping rule (paired with the PBGI or LogEIPC acquisition function) also achieves cost-adjusted simple regret no worse than a naive baseline—stopping-immediately (*Immediate*). Notably, this guarantee may not hold for other acquisition–stopping rule pairings. Moreover, in the worst case, this is the best guarantee we can hope for—for instance, the evaluation costs can be uniformly high and no point is worth evaluating.

**Theorem 2.** *Consider the setting and algorithm specified in Lemma 1. Let $U := \mu(x_1) - \mathbb{E}[\min_{x \in X} f(x)] < \infty$, then the algorithm's expected cost-adjusted simple regret is bounded by*

$$\mathbb{E}\left[ y_{1:\tau}^* - \min_{x \in X} f(x) + \sum_{t=1}^{\tau} c(x_t) \right] \leq \mathbb{E}\big[ \underbrace{y_1 - \min_{x \in X} f(x) + c(x_1)}_{\text{cost-adjusted regret of immediate stopping}} \big] = U + C. \tag{11}$$

---

[5]In fact, any acquisition function (e.g., expected improvement-cost (EIC) (Hu et al., 2025)) that ensures the one-step expected improvement is worth the evaluation cost before the stopping time $\tau$ can apply here.

*Proof.* We treat the two algorithms—meaning, the two acquisition function and stopping rule pairs—together and write $\mathcal{F}_t = \sigma(\{x_i, y_i\}_{i=1}^t)$ for the filtration generated by the observations. Since we are minimizing, the one–step improvement after iteration $t$ is $y_{1:t-1}^* - y_{1:t}^*$, where recall that $y_{1:t}^* = \min_{1 \le i \le t} y_i$. By our assumption about the random function $f$, $\mathbb{E}[y_{1:1}^*] = \mu(x_1)$ and the quantity $y_{1:1}^* - \min_{x \in X} f(x)$ has finite expectation $U < \infty$. Denote the posterior expected improvement function as $\alpha_t^{\mathrm{EI}}(x) = \mathrm{EI}_{f|x_{1:t}, y_{1:t}}(x; y_{1:t}^*)$.

By Lemma 1, $c(x_t) \le \alpha_{t-1}^{\mathrm{EI}}(x_t)$ for all $t \le \tau$. Set $\Delta_t = y_{1:t-1}^* - y_{1:t}^* \ge 0$ for $2 \le t \le \tau$. Conditional on $\mathcal{F}_{t-1}$ and the choice of $x_t$, we have

$$\mathbb{E}[\Delta_t \mid \mathcal{F}_{t-1}, x_t] = \alpha_{t-1}^{\mathrm{EI}}(x_t). \tag{15}$$

Taking expectations and summing up from 2 to $\tau$,

$$\mathbb{E}\left[\sum_{t=2}^{\tau} \alpha_{t-1}^{\mathrm{EI}}(x_t)\right] = \mathbb{E}\left[\sum_{t=2}^{\infty} \mathbf{1}_{\{t \le \tau\}} \alpha_{t-1}^{\mathrm{EI}}(x_t)\right]$$

$$= \mathbb{E}\left[\sum_{t=2}^{\infty} \mathbf{1}_{\{t \le \tau\}} \mathbb{E}[\Delta_t \mid \mathcal{F}_{t-1}, x_t]\right]$$

$$= \mathbb{E}\left[\sum_{t=2}^{\infty} \mathbf{1}_{\{t \le \tau\}} \Delta_t\right] \quad \text{(tower property)}$$

$$= \mathbb{E}\left[\sum_{t=2}^{\tau} \Delta_t\right] = \mathbb{E}[y_{1:1}^* - y_{1:\tau}^*]. \tag{16}$$

Summing (14) over $t \le \tau$, taking expectations, and applying (16), we have

$$\mathbb{E}\left[\sum_{t=2}^{\tau} c(x_t)\right] \le \mathbb{E}\left[\sum_{t=2}^{\tau} \alpha_{t-1}^{\mathrm{EI}}(x_t)\right] \le \mathbb{E}[y_{1:1}^* - y_{1:\tau}^*]. \tag{17}$$

Adding the term $y_{1:\tau}^*$ on both sides of (17) gives

$$\mathbb{E}\left[y_{1:\tau}^* + \sum_{t=2}^{\tau} c(x_t)\right] \le \mathbb{E}[y_{1:1}^*].$$

Finally, adding $c(x_1)$ to both sides and subtracting $\mathbb{E}[\min_{x \in X} f(x)]$ yields

$$\mathbb{E}\left[y_{1:\tau}^* - \min_{x \in X} f(x) + \sum_{t=1}^{\tau} c(x_t)\right] \le \mathbb{E}\left[y_{1:1}^* - \min_{x \in X} f(x) + c(x_1)\right]$$

$$= \mathbb{E}\left[y_1 - \min_{x \in X} f(x) + c(x_1)\right]$$

$$= \mu(x_1) - \mathbb{E}\left[\min_{x \in X} f(x)\right] + \mathbb{E}[c(x_1)]$$

$$= U + C,$$

since $\mathbb{E}[y_1] = \mu(x_1)$, $C = c(x_1)$, and $U = \mu(x_1) - \mathbb{E}[\min_{x \in X} f(x)]$. This is exactly (11). $\square$

**Remark 5.** *The quantities $U$ and $C$ depend on the choice of the initial point $x_1$. Choosing $x_1$ to minimize $\mu(x_1) + c(x_1)$ yields a tighter bound.*

**Remark 6.** *In practice, one may center the objective by replacing $f(x)$ with $f(x) - \mu(x)$ and then using a zero-mean GP prior. This changes only the parameterization of the posterior mean (which can be shifted back by $\mu(x)$) and does not affect the implementation of the acquisition functions or stopping rules.*

**Corollary 3.** *Consider the setting and algorithm specified in Lemma 1. Then the expected cumulative cost of the algorithm is bounded by $\mathbb{E}\left[\sum_{t=1}^{\tau} c(x_t)\right] \le U + C$. Further, if evaluation costs are uniformly bounded below by a constant $c_0 > 0$, i.e., $c(x) \ge c_0, \forall x \in X$, then for any $\delta \in (0, 1)$, the algorithm terminates in at most $\frac{U+C}{\delta \cdot c_0}$ iterations with probability $1 - \delta$.*

*Proof.* Immediately by Theorem 2 and the fact that $y_{1:\tau}^* - \min_{x \in X} f(x) > 0$ pointwise, we have that the expected cumulative cost of the algorithm

$$\mathbb{E}\left[\sum_{t=1}^{\tau} c(x_t)\right] \leq U + C. \tag{18}$$

By Markov's inequality, for any $\delta \in (0, 1)$,

$$\Pr\left[\sum_{t=1}^{\tau} c(x_t) \leq \frac{U+C}{\delta}\right] \geq 1 - \delta.$$

Since $c(x_t) \geq c_0$ for all $t < \tau$, it follows that

$$\tau = \sum_{t=1}^{\tau} 1 \leq \sum_{t=1}^{\tau} \frac{c(x_t)}{c_0} = \frac{1}{c_0} \sum_{t=1}^{\tau} c(x_t).$$

Therefore,

$$\Pr\left[\tau \leq \frac{U+C}{\delta \cdot c_0}\right] \geq \Pr\left[\frac{1}{c_0} \sum_{t=1}^{\tau} c(x_t) \leq \frac{U+C}{\delta \cdot c_0}\right] = \Pr\left[\sum_{t=1}^{\tau} c(x_t) \leq \frac{U+C}{\delta}\right] \geq 1 - \delta.$$

$\square$

**Corollary 4.** *Consider the setting, algorithm, and notation specified in Lemma 1 and Theorem 2, but with costs rescaled by a factor $\lambda > 0$: both the acquisition values and the stopping conditions are computed using $\lambda c(\cdot)$. If the cost-scaling factor is set to $\lambda = \frac{U}{B-C}$, then the algorithm's expected cumulative unscaled cost satisfies $\mathbb{E}[\sum_{t=1}^{\tau} c(x_t)] \leq B$.*

*Proof.* Since the Bayesian optimization algorithm considers the post-scaling cost, by Theorem 2 and the fact that $y_{1:\tau}^* - \min_{x \in X} f(x) > 0$ pointwise, we have

$$\mathbb{E}\left[\sum_{t=1}^{\tau} \lambda \cdot c(x_t)\right] \leq \mathbb{E}\left[y_{1:\tau}^* - \min_{x \in X} f(x) + \sum_{t=1}^{\tau} \lambda \cdot c(x_t)\right] \leq \mu(x_1) - \mathbb{E}\left[\min_{x \in X} f(x)\right] + \lambda C = U + \lambda C. \tag{19}$$

Since $\lambda = U/(B - C)$, we have

$$\mathbb{E}\left[\sum_{t=1}^{\tau} c(x_t)\right] \leq \frac{U + \lambda C}{\lambda} = \frac{U}{\lambda} + C = B.$$

$\square$

**Remark 7.** *In this paper, we focus on the setting where $f$ is drawn from a Gaussian process, i.e., $f \sim \mathcal{GP}(\mu(\cdot), K)$. In this case, the term $U = \mu(x_1) - \mathbb{E}[\min_{x \in X} f(x)]$ can be further bounded above using classical results on the expected supremum/infimum of Gaussian processes; see, for example, Lifshits (2012, Theorem 10.1).*

## C   EXPERIMENTAL SETUP

All experiments are implemented based on BoTorch (Balandat et al., 2020). Each Bayesian optimization procedure is initialized with $2(d + 1)$ random samples, where $d$ is the dimension of the search domain. For Bayesian regret experiments, we follow the standard practice to generate the initial random samples using a quasirandom Sobol sequence. For empirical experiments, we randomly sample configuration IDs from a fixed pool of candidates—2,000 for LCBench and 32,768 for NATS-Bench. All computations are performed on CPU.

Each experiment is repeated with 50 random seeds, and we report the mean with error bars, given by two times the standard error, for each stopping rule. We also impose a cap on the number of iterations: 100 for 1D Bayesian regret, 500 for 8D Bayesian regret, and 200 for empirical experiments. If a stopping rule is not triggered before reaching this cap, we treat the stopping time as equal to the cap.

**Gaussian process models.** For all experiments, we follow the standard practice to apply Matérn kernels with smoothness $5/2$ and length scales learned from data via maximum marginal likelihood optimization, and standardize input variables to be in $[0,1]^d$. For empirical experiments, we standardize output variables to be zero-mean and unit-variance, but not for Bayesian regret experiments. In this work, we consider the noiseless setting and set the fixed noise level to be $10^{-6}$. In the unknown-cost experiments, we follow Astudillo et al. (2021) to model the objective and the logarithm of the cost function using independent Gaussian processes.

**Acquisition function optimization.** For the 1D Bayesian regret experiments, we optimize over 10,001 grid points. For the 8D Bayesian regret experiment, we use BoTorch's 'gen_candidates_torch', a gradient-based optimizer for continuous acquisition function maximization, as it avoids reproducibility issues caused by internal randomness in the default scipy optimizer. For the empirical experiments, since LCBench and NATS-Bench provide only 2,000 and 32,768 configurations respectively, we optimize the acquisition function by simply applying an argmin/argmax over the acquisition values of the unevaluated configurations, without using any gradient-based methods.

**Acquisition function and stopping rule parameters.** For PBGI, we follow Xie et al. (2024) to compute the Gittins indices using 100 iterations of bisection search without any early stopping or other performance and reliability optimizations.

For UCB/LCB-based acquisition functions and stopping rules, we follow the original GP-UCB paper Srinivas et al. (2009) and the choice in UCB/LCB based stopping rules (Makarova et al., 2022; Ishibashi et al., 2023) to use the schedule $\beta_t = 2\log(dt^2\pi^2/6\delta)$, where $d$ is the dimension. We also adopt their choice of $\delta = 10^{-1}$ and a scale-down factor of 5.

For SRGap-med, which stops when the simple regret gap falls below $\chi$ times the median of its initial $T = 20$ values, we set $\chi = 0.1$ in the empirical experiments, instead of the default value $\chi = 0.01$ recommended in the literature. This adjustment was made because SRGap-med tends to stop too late on the LCBench datasets, likely due to the relatively small initial regret values and insignificant drop over time.

For PRB, we follow Wilson (2024) to use the schedule $N_t = \max(\lceil 64 * 1.5^{t-1}\rceil, 1000)$ for number of posterior samples, risk tolerance $\delta = 0.05$. The error bound $\epsilon$ is set to be 0.1 for Bayesian regret experiments and 0.5% of the best test error (here, the misclassification rate) among all configurations for the empirical experiments.

For the 8D Bayesian regret experiments, for all acquisition-value-based stopping rules, we apply moving average over 20 iterations to mitigate the fluctuations due to the imperfect acquisition function optimization. Figure 4 illustrates the challenges these oscillations pose when computing stopping rule statistics and shows the improvement with moving average. For consistency, we also apply the 20-iteration averaging to the non-acquisition-value-based GSS and Convergence baselines.

**Omitted baselines.** We omit the KG acquisition function and stopping rule due to its high computational cost, as they are shown to be computationally intensive in the runtime experiments of Xie et al. (2024).

**Objective functions: Bayesian regret.** In all Bayesian regret experiments, each objective function $f$ is sampled from a Gaussian process prior with a Matérn 5/2 kernel and a length scale of $0.1$, using a different seed in each of the 50 trials.

**Objective functions: empirical.** In the empirical experiments, the validation error (scaled out of 100) is used as the objective function during the Bayesian optimization procedure, while the cost-adjusted simple regret is reported based on the corresponding test error.

**Cost functions: Bayesian regret.** In Bayesian regret experiments, we consider three types of evaluation costs: uniform, linear, and periodic. These costs are normalized such that $\mathbb{E}_{x\in[0,1]^d}[c(x)]$ is approximately 1 and their expressions (prior to cost scaling) are given below.

In the *uniform* cost setting, each evaluation incurs a constant cost of 1.

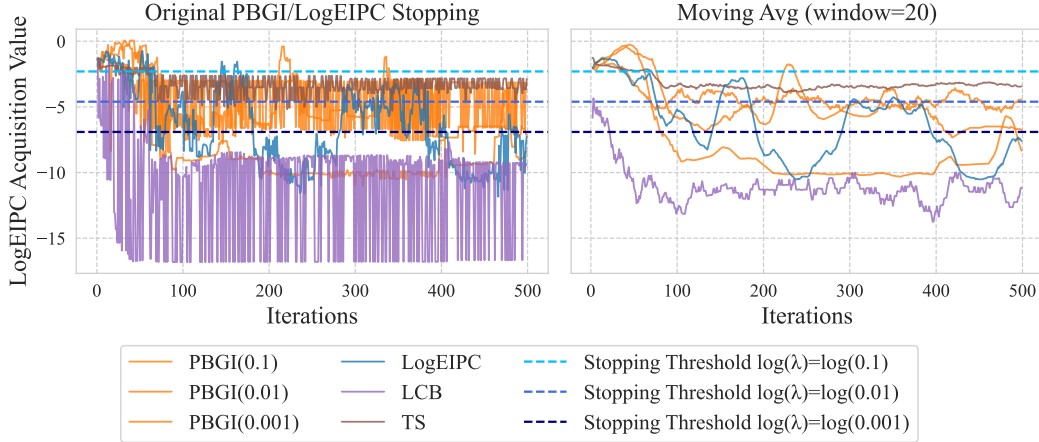

Figure 4: Comparison of the raw and moving-averaged PBGI/LogEIPC stopping rule signals (i.e., the LogEIPC acquisition values) in Bayesian regret 8D experiments for multiple acquisition functions, with linear cost and stopping thresholds at $\log(0.1)$, $\log(0.01)$ and $\log(0.001)$. *Left:* The unaveraged signals exhibits large, high-frequency fluctuations due to the difficulty of acquisition optimization in high dimensions. *Right:* Applying moving average (window=20) smooths these wiggles, yielding more stable stopping signals.

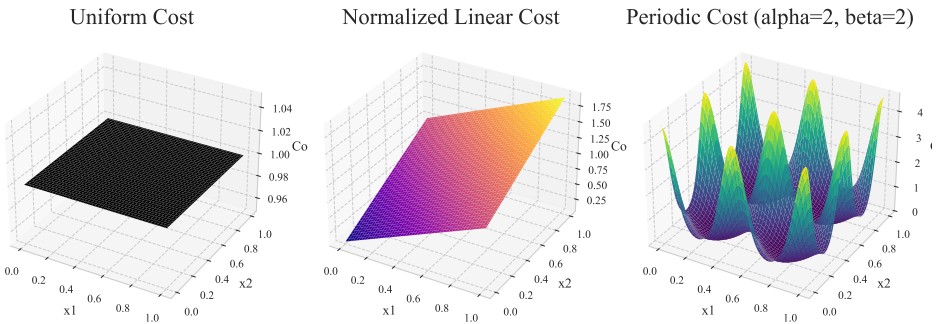

Figure 5: Surface plots of cost functions over $[0,1]^2$: (Left) Uniform cost of 1 across domain. (Middle) Normalized linear cost increasing with the mean of $x_1$ and $x_2$. (Right) Periodic cost with $\alpha = 2$, $\beta = 2$, normalized by Bessel-based factor.

In the *linear* cost setting, the cost increases proportionally with the average coordinate value of the input:

$$\text{linear\_cost}(x) = \frac{1 + 20 \cdot \left(\frac{1}{d} \sum_{i=1}^{d} x_i\right)}{11}.$$

In the *periodic* cost setting, the evaluation cost fluctuates across the domain. Following Astudillo et al. (2021), we define the periodic cost as

$$\text{periodic\_cost}(x) = \frac{\exp\left(\frac{\alpha}{d} \sum_{i=1}^{d} \cos\left(2\pi\beta(x_i - x_i^*)\right)\right)}{\left[I_0\left(\frac{\alpha}{d}\right)\right]^d},$$

where $x_i^*$ denotes the coordinate of the global optimum of $f$, and $I_0$ is the modified Bessel function of the first kind, which acts as a normalization constant. We set $\alpha = 2$ and $\beta = 2$ to induce noticeable variation in cost across the domain, while ensuring that costly evaluations can still be worthwhile.

A visualization of the three cost functions is provided in Figure 5.

**Cost functions: empirical.** In the *unknown-cost* experiments, we treat runtime—meaning, the provided full model training time (200 epochs for LCBench and 90 epochs for NATS)—as evaluation costs (prior to cost scaling).

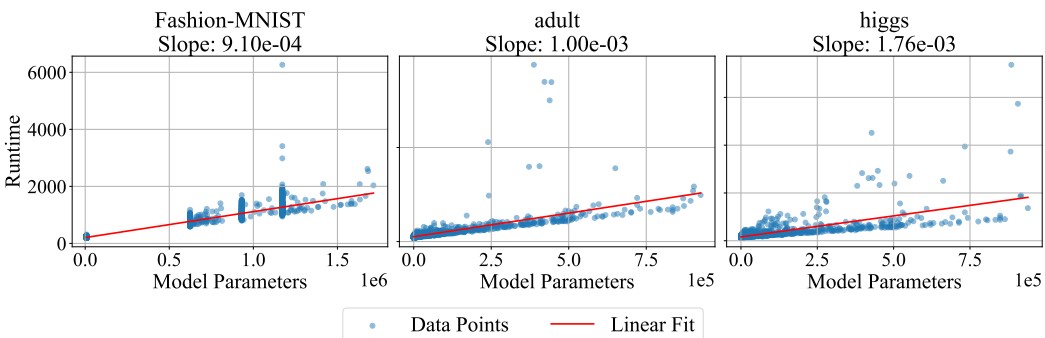

Figure 6: Empirical relationship between the number of model parameters and runtime for three LCBench datasets. Each subplot shows a scatter plot of actual runtime ($y$-axis) against number of model parameters ($x$-axis), along with a fitted linear regression line. The observed linear trend supports using 0.001 times the number of model parameters as a proxy for runtime. For *Fashion-MNIST* and *adult*, the fitted slopes are close to 0.001. The slope for higgs is slightly higher, possibly due to a few outliers.

In the *known-cost* experiments, for LCBench, we estimate the runtime cost from the number of model parameters $p$. Specifically, for the first three datasets in the six-dataset lite version, we use 0.001 times the number of model parameters as a proxy for runtime. This proxy is motivated by our observation of an approximately linear relationship between the number of model parameters and the actual runtime, with slope close to 0.001 (see Figure 6). For the full 35-dataset version, where the slope varies across datasets, we instead use a regression-derived coefficient $\alpha$ and the proxy cost is $\alpha p$. Importantly, the number of model parameters can be computed in advance, before the Bayesian optimization procedure, based on the network structure and classification task, as we explain in detail below.

In a feedforward neural network like shapedmlpnet with shape 'funnel', the model parameters are determined by input size (number of features), output size (e.g., number of classes), number of layers, size of each layer. The input size and output size are given by:

- Fashion-MNIST:
  - Input dimension: 784 (each image has 28×28 pixels, flattened into a vector of length 784)
  - Number of Output Features (output_feat): 10 (corresponding to 10 clothing categories)
- Adult:
  - Input Dimension: 14 (the dataset comprises 14 features, including both numerical and categorical attributes)
  - Number of Output Features: 1 (binary classification: income $> 50K$ or $\leq 50K$)
- Higgs:
  - Input Dimension: 28 (each instance has 28 numerical features)
  - Number of Output Features: 1 (binary classification: signal or background process)

The number of layers (num_layers) and size of each layer (max_unit) can be obtained from the configuration. With these information, we can compute the total number of model parameters (weights and biases) based on the layer-wise structure as follows:

$$\text{layer\_params}_{i \to i+1} = \text{layer}_i \cdot \text{layer}_{i+1} + \text{layer}_{i+1} \quad \text{(weights + biases)} \tag{20}$$

$$\text{model\_params} = \sum_{i=0}^{L-1} \text{layer\_params}_{i \to i+1} \tag{21}$$

$$\text{where} \quad \text{layer}_0 = \text{input\_dim}, \quad \text{layer}_L = \text{output\_feat} \tag{22}$$

Similarly for NATS-Bench, we use $\alpha \times F + \beta$ as a proxy for runtime (see Figure 7 for a visualization of the linear relationship), where $F$ is the number of floating point operations (FLOPs). Specifically, for

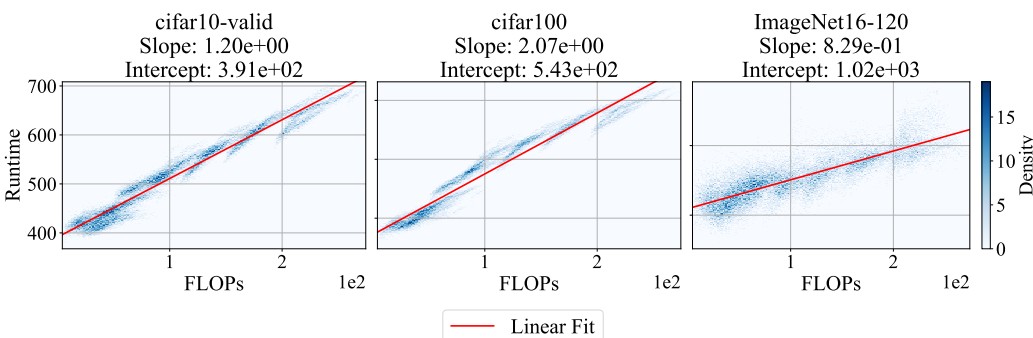

Figure 7: Empirical relationship between the number of FLOPs and runtime for the three NATS-Bench datasets. Each subplot shows a heatmap of actual runtime ($y$-axis) against number of FLOPs ($x$-axis), along with a fitted linear regression line. The observed linear trend supports using $\alpha \times \#\text{FLOPs} + \beta$ as a proxy for runtime.

*cifar10-valid*, we set $\alpha = 1$, $\beta = 400$; for *cifar100*, we set $\alpha = 2$, $\beta = 550$; and for *ImageNet16-120*, we set $\alpha = 1$, $\beta = 1000$.

FLOPs can also be computed in advance, as it is determined solely by the architecture's structure and the fixed input shape. Specifically, they are precomputed and stored for each architecture. Since each architecture corresponds to a deterministic computational graph and all inputs (e.g., CIFAR-10 images) have a fixed shape, the FLOPs required for a forward pass can be calculated analytically—without executing the model on data.

## D  ADDITIONAL EXPERIMENT RESULTS

In this section, we present additional experimental results to further evaluate the performance of different acquisition-function–stopping-rule pairs across various settings. We also include alternative visualizations to aid interpretation of the results.

### D.1  RUNTIME COMPARISON

First, we compare the runtime between our PBGI/LogEIPC stopping rule with several baselines. We measure the CPU time (in seconds) of the computation of the stopping rule, excluding the acquisition function computation and optimization.

From results in Figure 8 we can see that our PBGI/LogEIPC stopping rule is roughly as efficient as SRGap-med and UCB–LCB. In contrast, PRB is significantly more time-consuming, as it involves optimizing over up to 1000 samples.

### D.2  ORDER OF STOPPING AND POSTERIOR UPDATES

Following the discussions in Section 3, we always compute our proposed stopping rule with respect to the optimal acquisition function value of the *next round*—namely, the one which is obtained after posterior updates have been performed. One could alternatively consider checking the stopping criteria *before* posterior updates, which is backward-looking rather than forward-looking. Figure 9 provides an empirical comparison between the two choices, showing that stopping *after* the posterior update leads to stronger empirical performance.

This suggests that the theoretical guarantee for the Gittins index policy in the correlated Pandora's Box setting by Gergatsouli & Tzamos (2023), which is based on the *before-posterior-update* stopping, could potentially be improved by adopting the *after-posterior-update* stopping.

### D.3  ADDITIONAL EXPERIMENT RESULTS: EMPIRICAL

This subsection presents additional results for hyperparameter optimization on the LCBench datasets and neural architecture–size search on the three NATS-Bench datasets. For LCBench, we provide

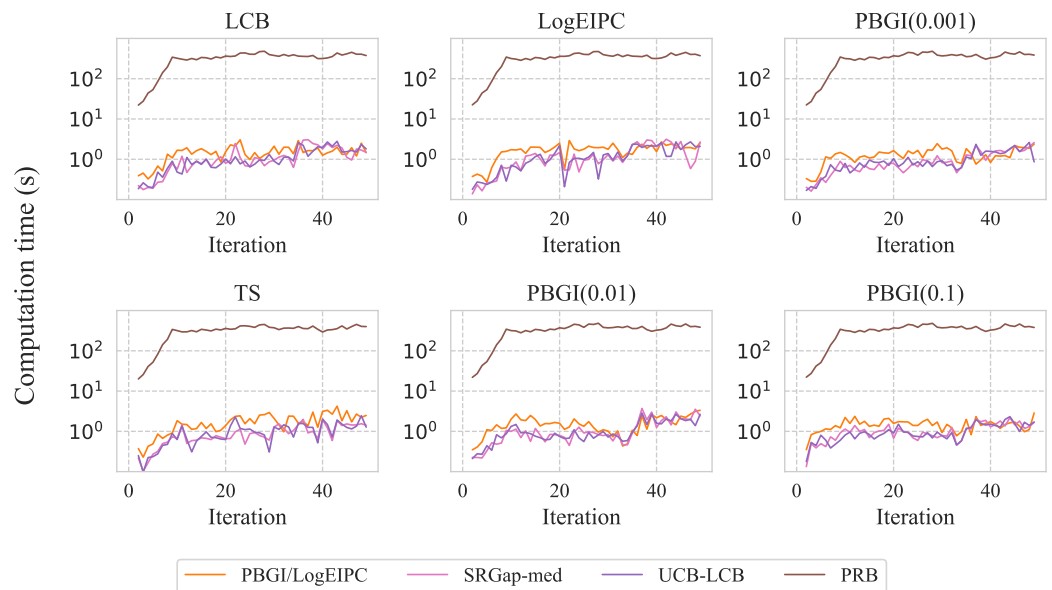

Figure 8: Evolution of per-iteration computation time (in log scale) for different stopping stopping rules when paired with six acquisition policies on the 8-dimensional Bayesian regret benchmark. Each subplot shows the average runtime (in seconds) over 50 iterations under one acquisition function—LCB, Thompson Sampling, LogEIPC, PBGI($\lambda = 10^{-1}$), PBGI($\lambda = 10^{-2}$), and PBGI($\lambda = 10^{-3}$). Curves correspond to four stopping criteria: our PBGI/LogEIPC stopping rule, SRGap-med, UCB–LCB, and the probabilistic regret bound (PRB). Convergence and GSS can be applied using only the best observed value and thus require no additional computation time, thus they are omitted here. LogEIPC-med relies on the same underlying statistical computations as the LogEIPC rule, and therefore its runtime is not measured separately. The results should that PRB incurs significant computational overhead compared to other stopping rules.

results for the first three datasets from the six-dataset lite version of the benchmark: Fashion-MNIST, adult, and higgs under alternative settings (fixed budget, actual runtime, and unknown cost), along with an alternative visualization. We also include per-dataset results under the proxy-runtime setting for the full set of 35 datasets.

**Simple regret under the fixed-budget setting.** To isolate the effect of the acquisition function on cost-adjusted regret, we report the simple regret of several acquisition functions in the fixed-budget setting. We compare LogEIPC, PBGI, LCB, and TS, and additionally include PBGI-D with our recommended choice $\lambda_0 = U/(B - C)$ from Section 3.1.1, where $U = 50$ (reflecting the [0,100] range of classification accuracy) and $B - C = 10,000$ (corresponding to a budget after initial evaluation of 10,000 seconds, or roughly 3 hours). Cost-aware methods (LogEIPC and the PBGI variants) outperform cost-unaware ones (UCB and TS), with PBGI at smaller $\lambda$ consistently better than LogEIPC. This mirrors the findings of Xie et al. (2024) and helps explain the strong performance of the matched PBGI combination in our main experiments.

**Number of trials where stopping fails.** We count the number of trials in which a stopping rule fails to trigger within our iteration cap of 200 and present the results in Table 1. From the table, we observe that on datasets from the NATS benchmark, regret-based and acquisition-based stopping rules—except for ours—often fail to stop early. On LCBench datasets, some regret-based stopping rules such as SRGap-med and UCB–LCB also frequently exceed the cap. In contrast, our stopping rule consistently stops early, which aligns with our theoretical result in Corollary 4.

**Alternative visualization: cost-adjusted regret vs iteration.** We provide an alternative visualization of cost-adjusted simple regret by plotting its mean and error bars at fixed iterations, along with the mean and error bars of the stopping iterations for each rule. This allows us to compare the

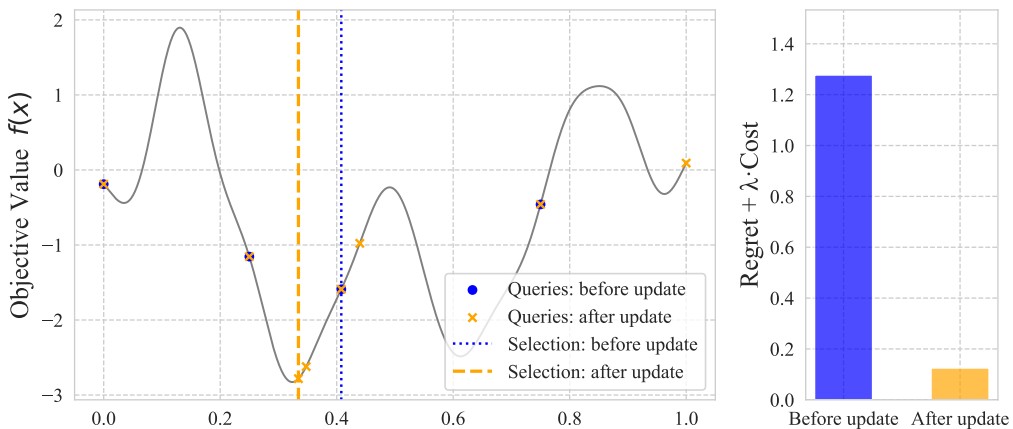

Figure 9: Illustration of a single draw from a Matérn-$5/2$ Gaussian process on $[0, 1]$ with lengthscale 0.1, optimized using PBGI acquisition function under uniform cost and cost-scaling factor $\lambda = 0.01$. We compare two variants of PBGI stopping rules: the *before-posterior-update* (this-round) stopping rule and the *after-posterior-update* (next-round) stopping rule. **Left:** The latent objective function (solid gray) and evaluation sequences for *before-posterior-update* stopping (blue circles) and *after-posterior-update* stopping (orange crosses). The dotted blue line and the dashed orange line mark the best observed value under each respective rule. **Right:** Cost-adjusted regret for each stopping rule. In this example, *after-posterior-update* stopping achieves strictly lower cost-adjusted regret despite performing more evaluations.

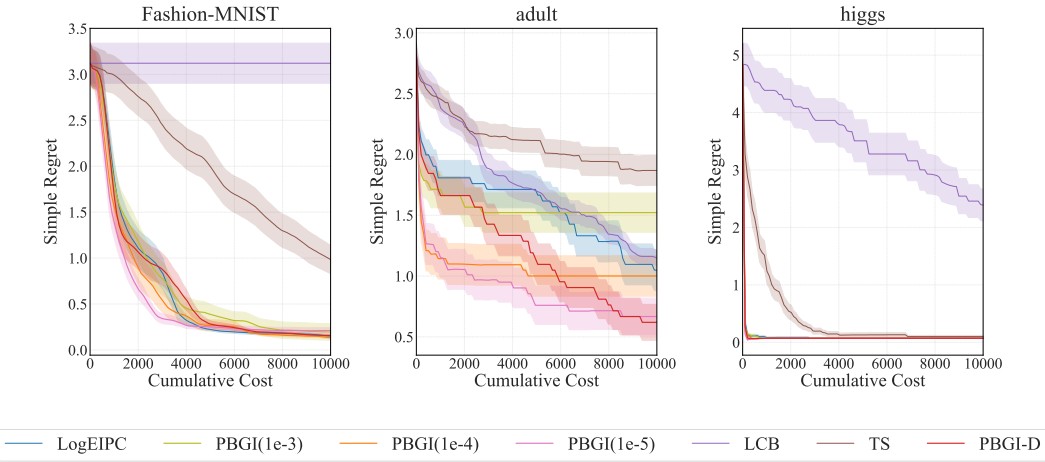

Figure 10: Comparison of simple regret of five acquisition functions: LogEIPC, PBGI($\lambda = 10^{-4}$), LCB, TS, and PBGI-D on LCBench datasets, using scaled proxy runtime as evaluation cost. We can see that indeed the cost-aware LogEIPC and PBGI outperforms the cost-unaware UCB and TS. PBGI-D with out recommended $\lambda_0$ is also competitive.

performance of adaptive stopping rules not only against the hindsight-optimal adaptive stopping but also against the hindsight-optimal fixed-iteration stopping.

As shown in the empirical setting in Figures 11 to 16, cost-adjusted regret generally decreases in the early iterations and then increases. The turning point is exactly the hindsight-optimal fixed-iteration stopping point, and our PBGI/LogEIPC stopping rule consistently performs close to this optimum, particularly when paired with the PBGI acquisition function.

**Cost model mismatch: proxy runtime vs actual runtime.** In the known-cost setting of hyper-parameter tuning, a practical approach is to use a proxy for runtime as the evaluation cost during the Bayesian optimization procedure. In our case, we use the number of model parameters scaled

Table 1: Number of trials (out of 50) where each stopping rule failed to trigger within 200 iterations, for each dataset in the LCBench (first three) and NATS (last three) benchmarks and each acquisition function. Results are identical across acquisition functions.

| Dataset | PBGI | LogEIPC-med | SRGap-med | UCB–LCB | GSS | Convergence | PRB |
|---|---|---|---|---|---|---|---|
| Fashion-MNIST | 0 | 0 | 0 | 50 | 0 | 0 | 0 |
| adult | 5 | 0 | 7 | 38 | 0 | 0 | 6 |
| higgs | 0 | 0 | 31 | 50 | 0 | 0 | 0 |
| Cifar10 | 0 | 32 | 50 | 50 | 0 | 0 | 26 |
| Cifar100 | 3 | 17 | 50 | 50 | 0 | 0 | 2 |
| ImageNet | 0 | 23 | 50 | 50 | 0 | 0 | 0 |

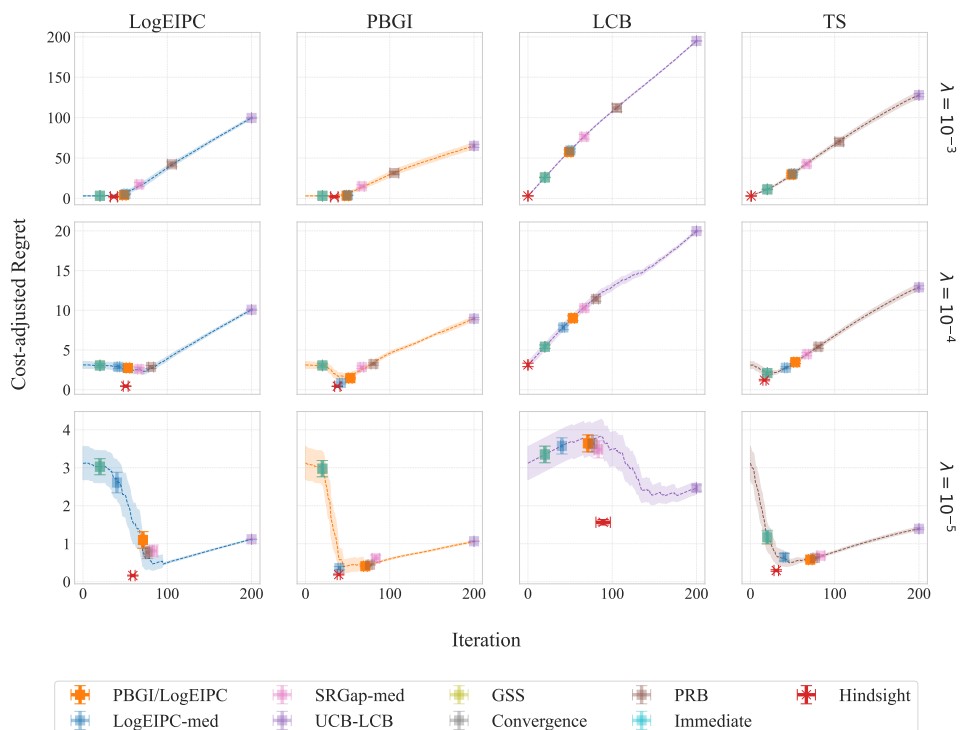

Figure 11: Comparison of cost-adjusted simple regret across acquisition function and stopping rule pairs on the *Fashion-MNIST* dataset. The objective function is the validation error, and the evaluation cost is the proxy runtime, scaled by three different cost-scaling factors $\lambda = 10^{-3}, 10^{-4}, 10^{-5}$. We can see that the PBGI/LogEIPC stopping rule consistently achieves cost-adjusted regret close to the hindsight optimal adaptive stopping as well as the hindsight optimal fixed-iteration stopping when paired with the PBGI acquisition function, though not always the best.

by a constant factor, which can be known in advance and has been shown to correlate well with the actual runtime. However, for reporting performance, one may prefer to use the actual runtime to better reflect real-world cost. To assess the impact of this cost model mismatch, we compare the cost-adjusted simple regret obtained when evaluation costs are computed using either the proxy runtime or the actual runtime. As shown in Figure 17, our PBGI/LogEIPC stopping rule remains close to the hindsight optimal even when there is a mismatch, although its ranking may shift slightly (e.g., from best to second-best on the *higgs* dataset).

**Unknown-cost.** Astudillo et al. (2021, Proposition 2) proposed modeling unknown cost $c(x)$ via

$$\mathbb{E}[1/c(x)]^{-1} = \exp(\mu_{\ln c}(x) - (\sigma_{\ln c}(x))^2/2). \tag{23}$$

An alternative is

$$\mathbb{E}[c(x)] = \exp(\mu_{\ln c}(x) + (\sigma_{\ln c}(x))^2/2). \tag{24}$$

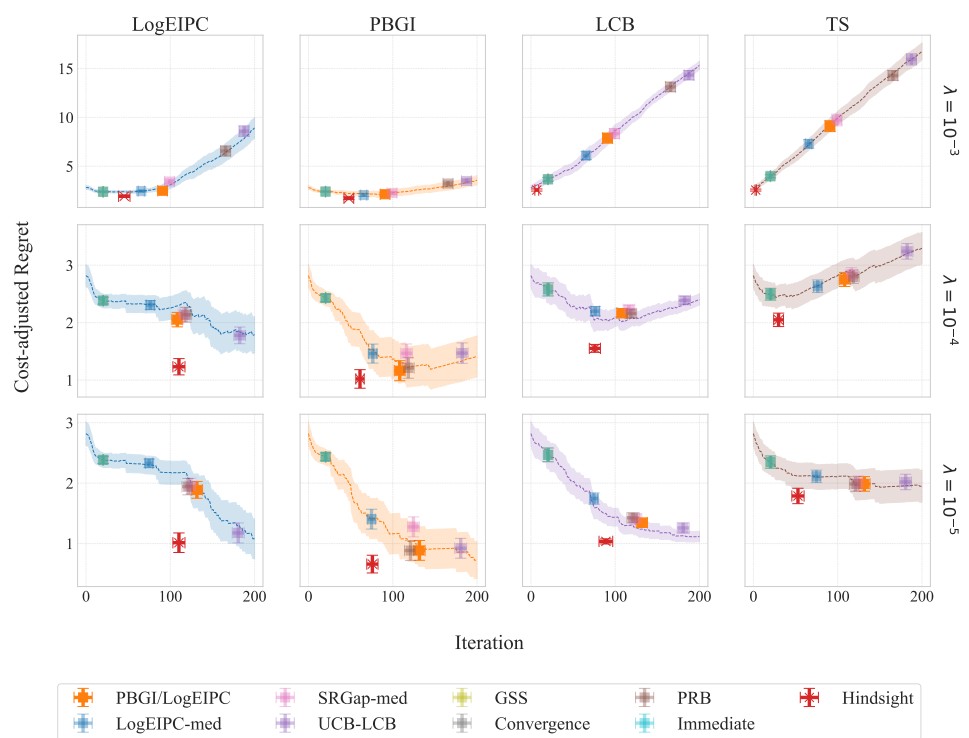

Figure 12: Comparison of cost-adjusted simple regret across acquisition function and stopping rule pairs on the *adult* dataset. The objective function is the validation error, and the evaluation cost is the proxy runtime, scaled by three different cost-scaling factors $\lambda = 10^{-3}, 10^{-4}, 10^{-5}$. We can see that the PBGI/LogEIPC stopping rule consistently achieves the cost-adjusted regret close to hindsight optimal adaptive stopping and hindsight optimal fixed-iteration stopping when paired with the PBGI or TS acquisition function, particularly with PBGI.

The difference in sign before the variance term reflects how each formulation handles predictive uncertainty: (23) encourages more exploration than (24). For PBGI under the unknown-cost setting, it is more natural to replace $c(x)$ in (5) with $\mathbb{E}[c(x)]$ using (24), as this aligns with how costs enter the root-finding problem. For LogEIPC, both variants are possible—we refer to the (23) version as *LogEIPC-inv* and the (24) version as *LogEIPC-exp*. However, equivalence between PBGI and LogEIPC stopping rules and our theoretical guarantees hold only with (24) but not with (23). Accordingly, we use (24) for both methods in our experiments to maintain consistency and preserve this equivalence. Figure 18 shows performance of acquisition function and stopping rule pairs under the unknown-cost setting, which are qualitatively similar to the known-cost setting.

**Cost-adjusted simple regret of all LCBench datasets.** Due to space constraints, for LCBench, Figure 3 in the main text reports min–max normalized cost-adjusted simple regret, defined as

$$\frac{r - r_{\min}}{r_{\max} - r_{\min}},$$

where $r$ denotes cost-adjusted regret of a given acquisition function–stopping rule pair, and $r_{\min}$ and $r_{\max}$ are the minimum (including the hindsight optimal) and maximum cost-adjusted regret across all pairs. This normalization enables aggregation across datasets with different regret scales.

Here we present the *unnormalized* bar-plot results for each LCBench dataset (OpenML dataset size in parentheses) under all three cost-scaling parameters in Figures 19 to 21. As discussed in the main text, our PBGI/LogEIPC stopping rule—when paired with either PBGI or LogEIPC—achieves competitive performance on roughly 75% of the 35 datasets. However, it performs consistently poorly across all three $\lambda$ values on the following tasks: Amazon_employee_access (32769), Australian (690), KDDCup09_appetency (50000), cnae-9 (1080), credit-g (1000), fabert (8237), and vehicle (846). Two additional datasets, mfeat-factors (2000) and jasmine (2984), show acceptable performance

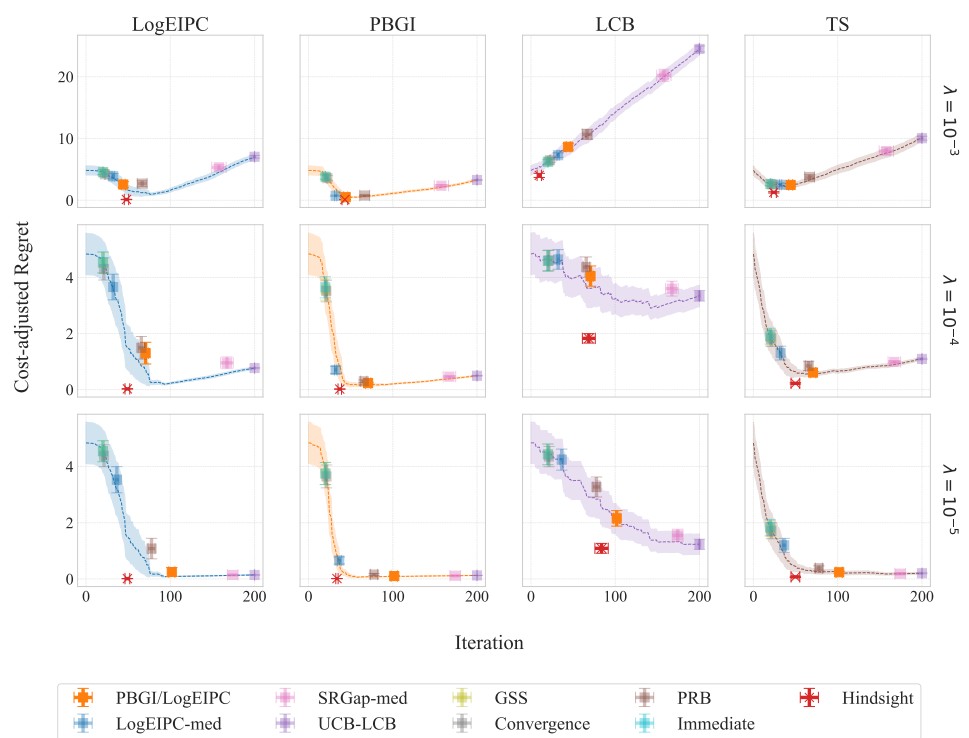

Figure 13: Comparison of cost-adjusted simple regret across acquisition function and stopping rule pairs on the *higgs* dataset. The objective function is the validation error, and the evaluation cost is the proxy runtime, scaled by three different cost-scaling factors $\lambda = 10^{-3}, 10^{-4}, 10^{-5}$. We can see that the PBGI/LogEIPC stopping rule consistently achieves the best cost-adjusted regret when paired with the LogEIPC, PBGI, or TS acquisition function, particularly with PBGI. These pairs not only approach the hindsight optimal adaptive stopping but also perform comparably to the hindsight optimal fixed-iteration stopping.

only when $\lambda = 10^{-5}$, and we conjecture that their relatively small dataset sizes (fewer than 10000 instances) may contribute to model misspecification and degraded performance at larger $\lambda$. The only exceptions to this size-related pattern are Amazon_employee_access and KDDCup09_appetency, whose poor performance cannot be explained solely by dataset size.

### D.4 ADDITIONAL EXPERIMENT RESULTS: BAYESIAN REGRET

In this section, we present the complete Bayesian regret results. Figures 22 to 24 show the 1D experiments, and Figures 25 to 27 show the 8D experiments. Each figure corresponds to one cost setting (uniform, linear or periodic) and three values of the cost-scaling factor, $\lambda = 10^{-1}, 10^{-2}, 10^{-3}$. In all of the experimental results, we observe that PBGI/LogEIPC acquisition function + PBGI/LogEIPC stopping achieves cost-adjusted regret that is not only competitive with the baselines, but is also competitive regarding the *best in hindsight* fixed iteration stopping and often competitive even comparing to *hindsight optimal* stopping. These results indicate that our automatic stopping rule can replace manual selection of stopping times without loss in performance. Figures 22 to 27 also show that our PBGI/LogEIPC stopping rule outperforms other baselines when the cost-scaling factor $\lambda$ is large, indicating that it's an especially suitable stopping criteria when evaluation is expensive.

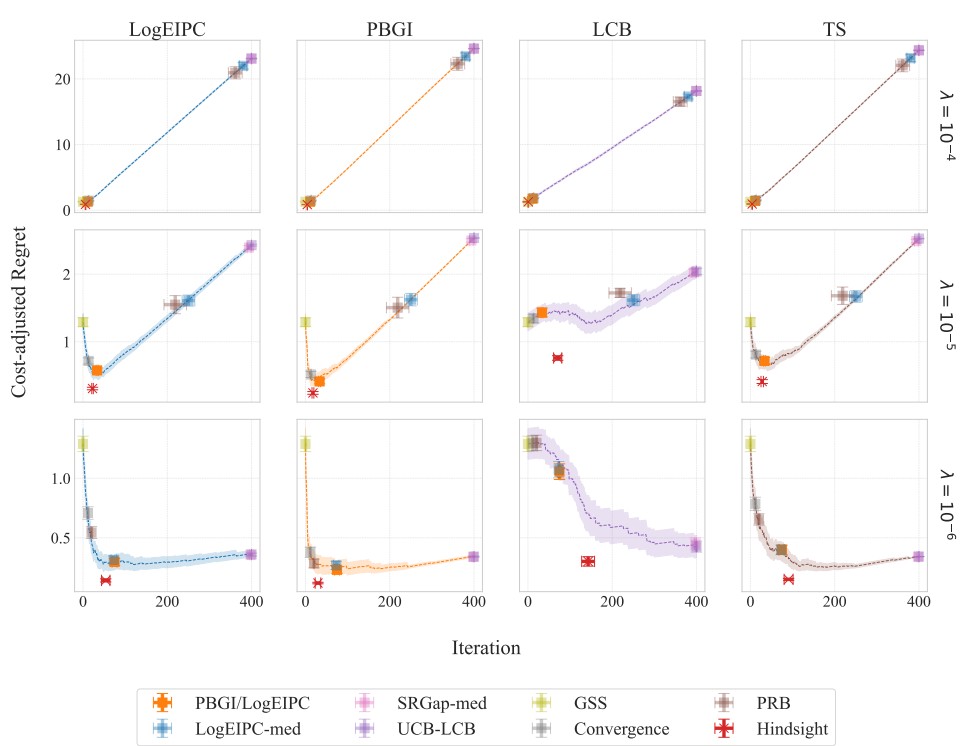

Figure 14: Comparison of cost-adjusted simple regret across acquisition function and stopping rule pairs on the *cifar10-valid* dataset. The objective function is the validation error, and the evaluation cost is the proxy runtime, scaled by three different cost-scaling factors $\lambda = 10^{-4}, 10^{-5}, 10^{-6}$. We can see that the PBGI/LogEIPC stopping rule consistently achieves the best cost-adjusted regret when paired with the LogEIPC, PBGI, or TS acquisition function, particularly with PBGI. These pairs not only approach the hindsight optimal adaptive stopping but also perform comparably to the hindsight optimal fixed-iteration stopping.

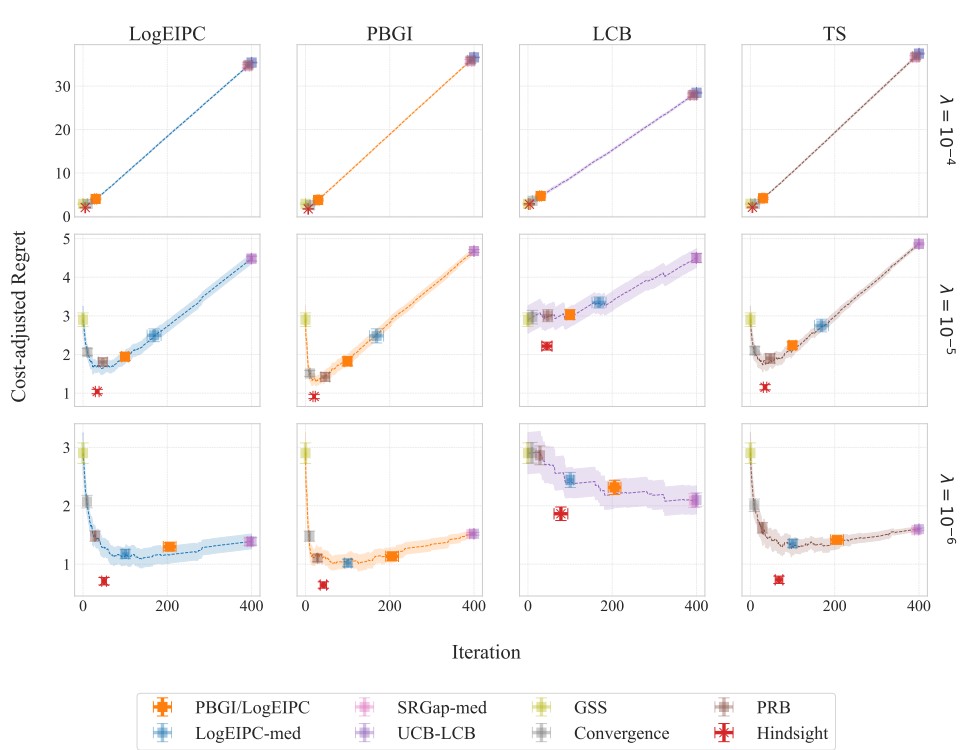

Figure 15: Comparison of cost-adjusted simple regret across acquisition function and stopping rule pairs on the *cifar100* dataset. The objective function is the validation error, and the evaluation cost is the proxy runtime, scaled by three different cost-scaling factors $\lambda = 10^{-4}, 10^{-5}, 10^{-6}$. The PBGI/LogEIPC stopping rule remains competitive at $\lambda = 10^{-4}$ and $10^{-6}$, though not always the best. At $\lambda = 10^{-5}$, unlike in most experiments, it stops noticeably late (though still outperforming several other rules), even when paired with its matching acquisition function. By Lemma 1, under model match, the PBGI/LogEIPC rule with the corresponding acquisition function should never incur negative expected cost-adjusted regret before stopping. The suboptimal behavior observed here points to significant model mismatch on the *cifar100* dataset.

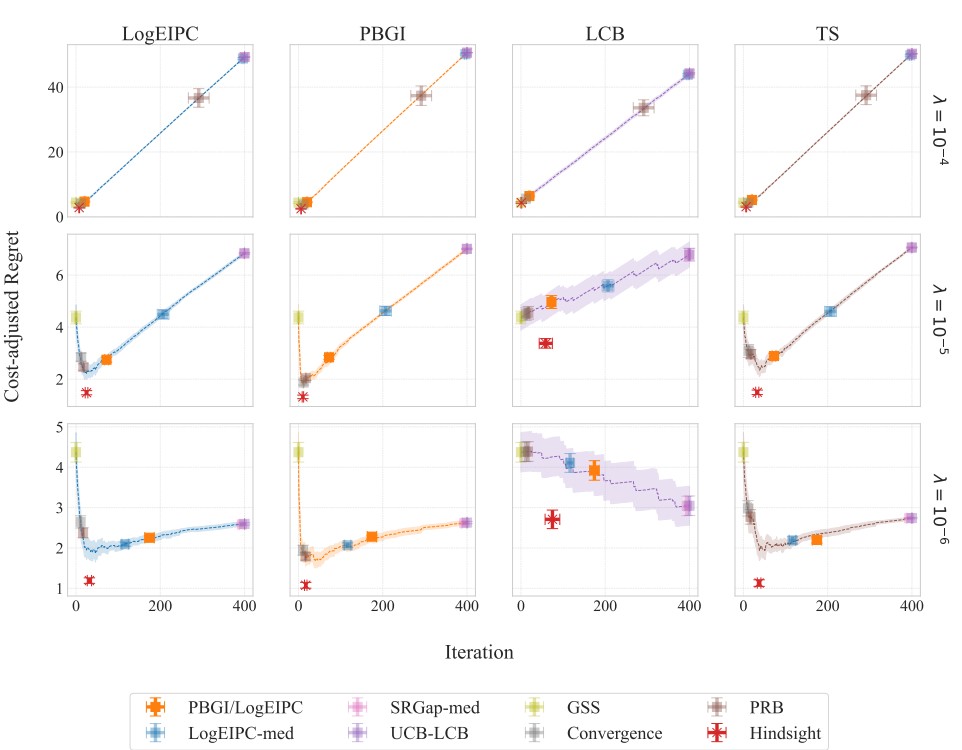

Figure 16: Comparison of cost-adjusted simple regret across acquisition function and stopping rule pairs on the *ImageNet16-120* dataset. The objective function is the validation error, and the evaluation cost is the proxy runtime, scaled by three different cost-scaling factors $\lambda = 10^{-4}, 10^{-5}, 10^{-6}$. The PBGI/LogEIPC stopping rule remains competitive at $\lambda = 10^{-4}$ and $10^{-6}$, though not always the best. At $\lambda = 10^{-5}$, unlike in most experiments, it stops noticeably late (though still outperforming several other rules), even when paired with its matching acquisition function. By Lemma 1, under model match, the PBGI/LogEIPC rule with the corresponding acquisition function should never incur negative expected cost-adjusted regret before stopping. The suboptimal behavior observed here points to significant model mismatch on the *ImageNet16-120* dataset.

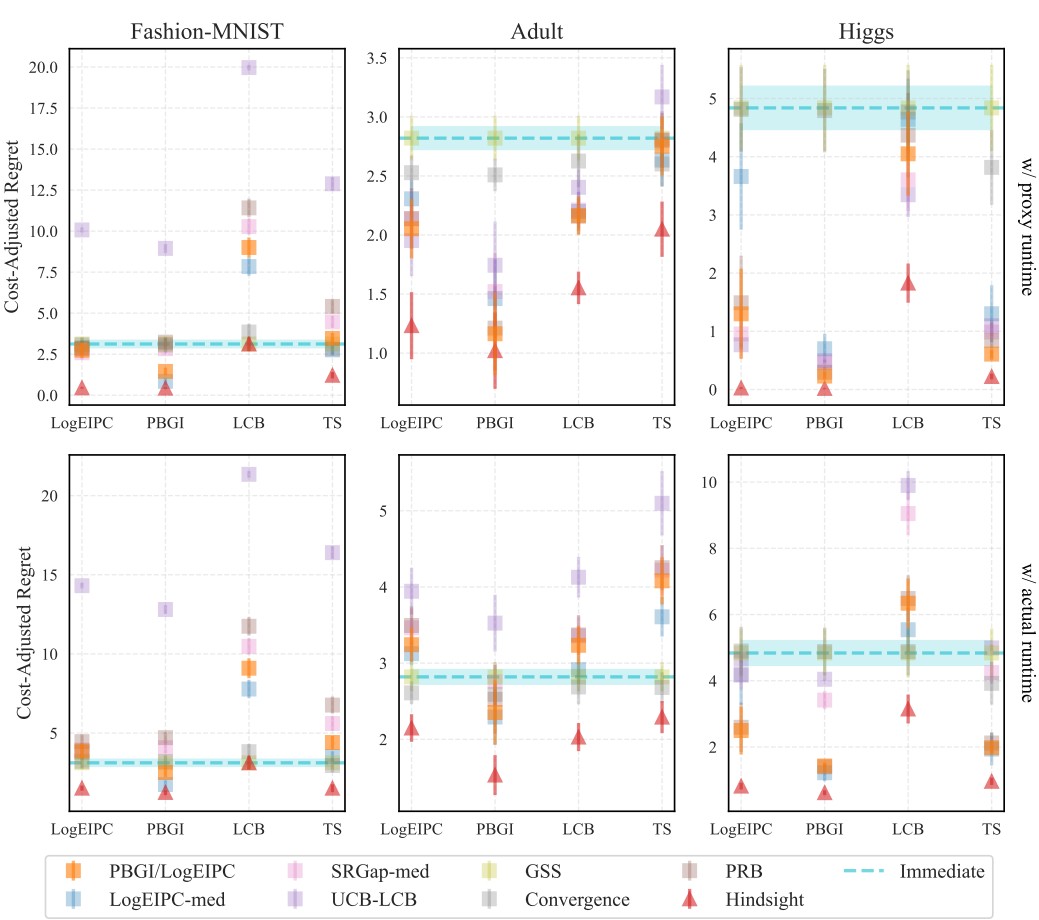

Figure 17: Comparison of cost-adjusted simple regret on three LCBench datasets with $\lambda = 10^{-4}$, using scaled proxy runtime vs. scaled actual runtime as evaluation cost. While our PBGI/LogEIPC stopping rule performs slightly worse under actual runtime (e.g., dropping from best to second-best on the *higgs* dataset), likely due to cost model mismatch, it remains close to the hindsight optimal in all cases.

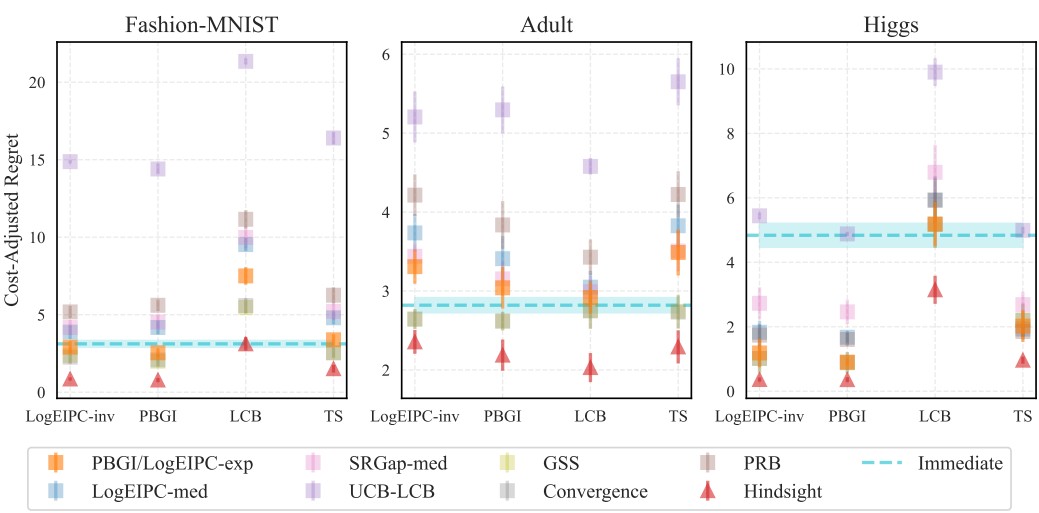

Figure 18: Cost-adjusted simple regret across acquisition function and stopping rule pairs under the unknown-cost setting on LCBench with $\lambda = 10^{-4}$. Our PBGI/LogEIPC stopping rule remains close to the hindsight optimal when paired with the LogEIPC-exp or PBGI acquisition function, sometimes slightly worse than heuristics such as GSS and Convergence. It is also slightly worse than Immediate on *Adult*, likely due to a cost-model mismatch in the unknown-cost setting.

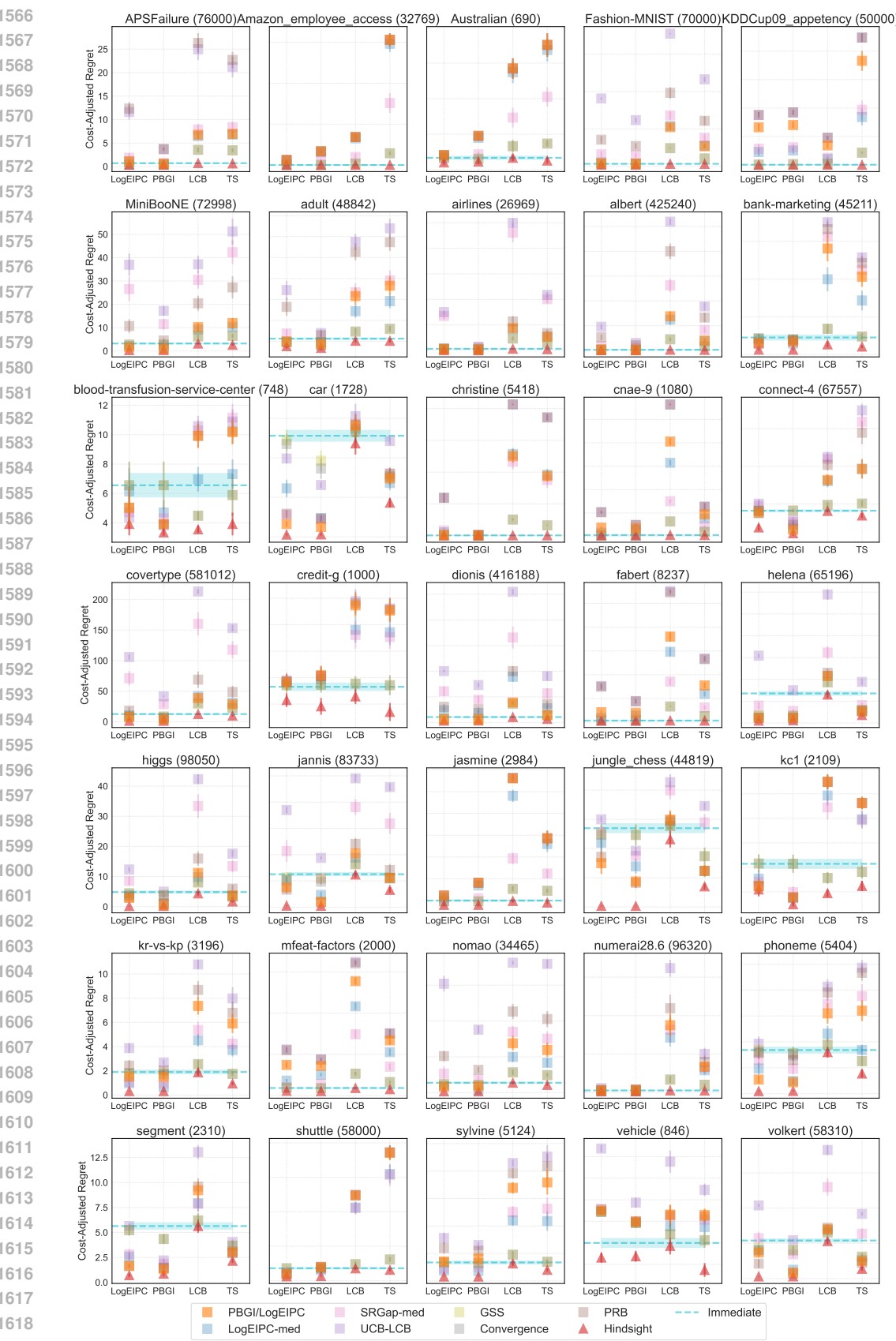

Figure 19: Cost-adjusted simple regret of all 35 LCBench datasets when $\lambda = 10^{-3}$.

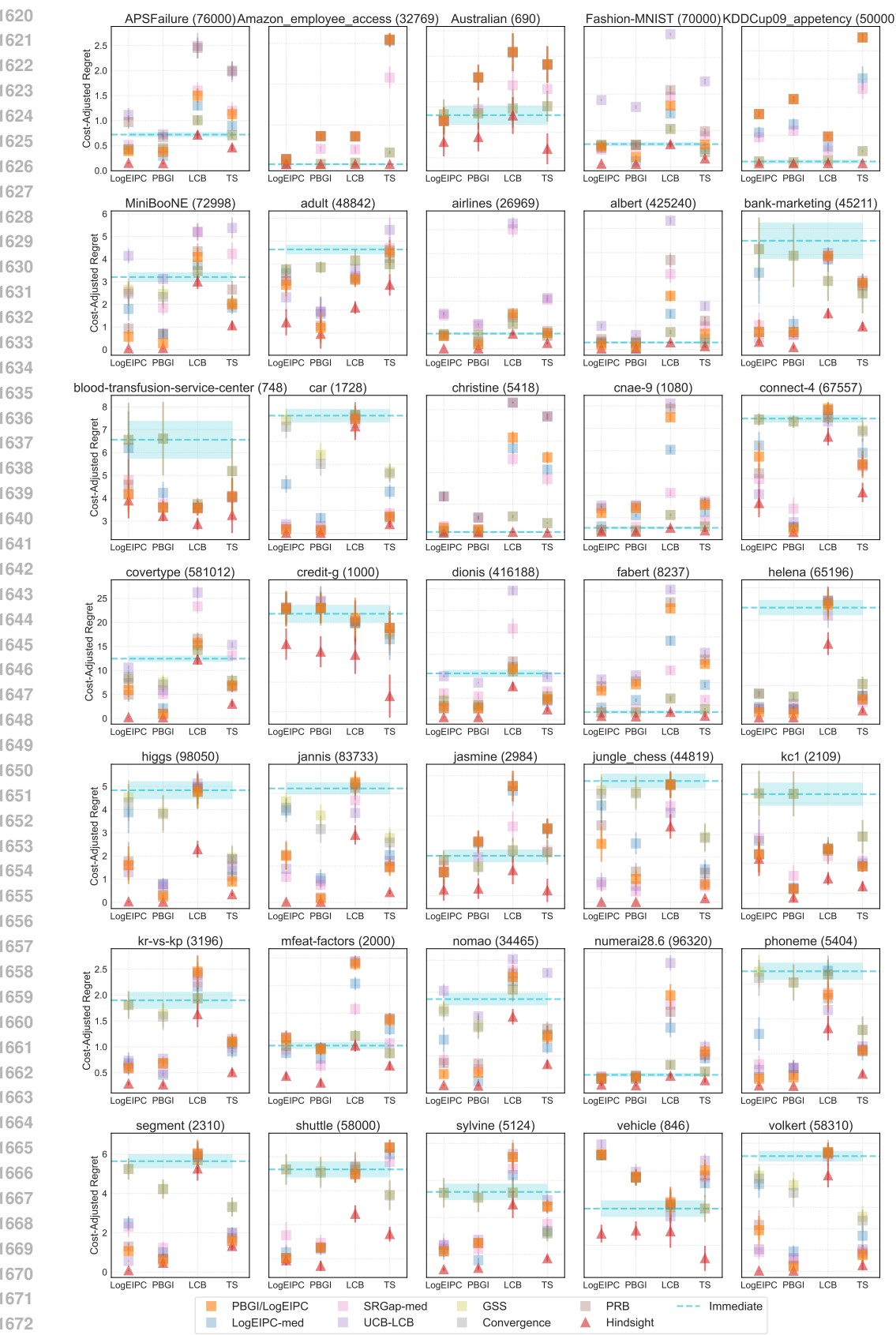

Figure 20: Cost-adjusted simple regret of all 35 LCBench datasets when $\lambda = 10^{-4}$.

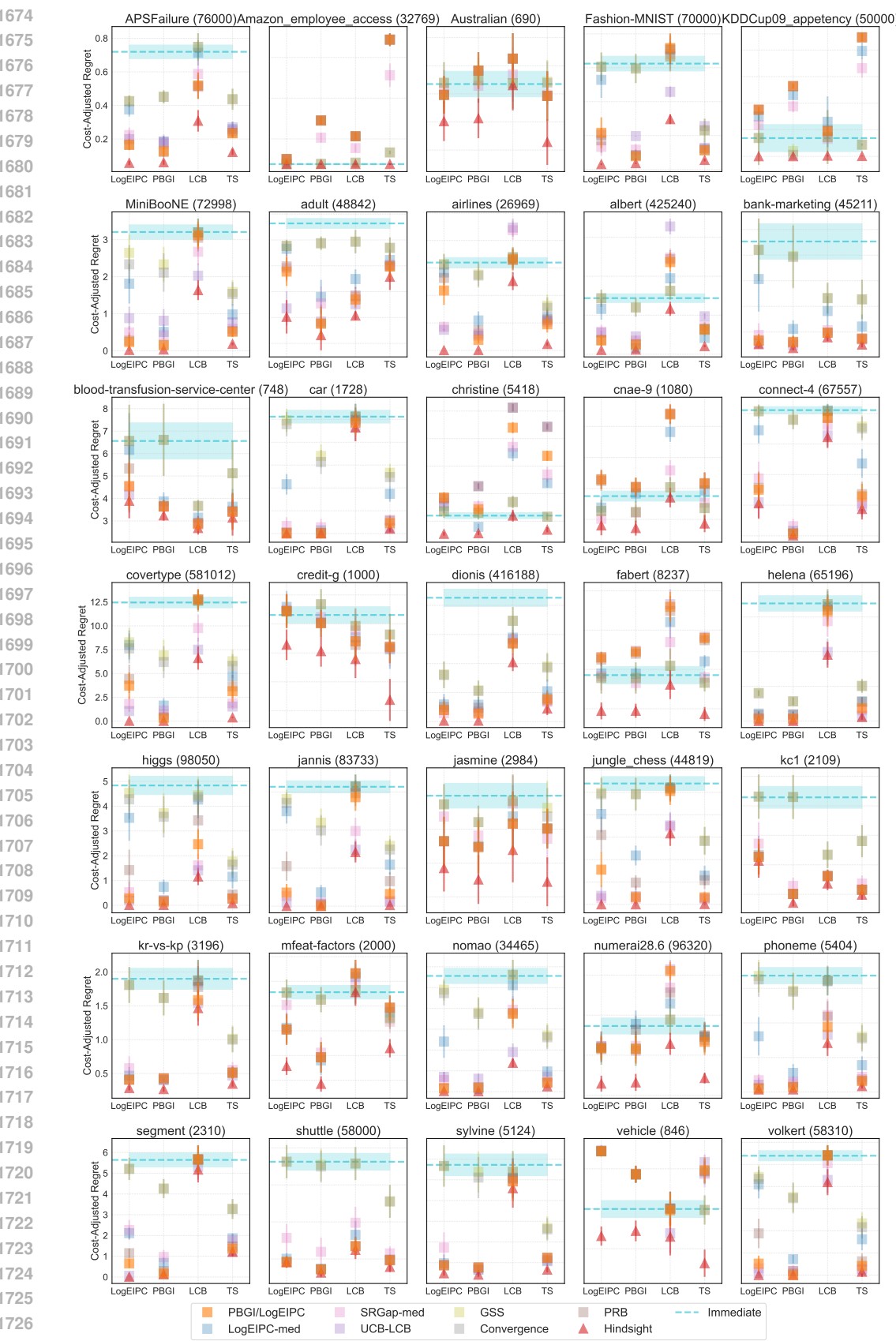

Figure 21: Cost-adjusted simple regret of all 35 LCBench datasets when $\lambda = 10^{-5}$.

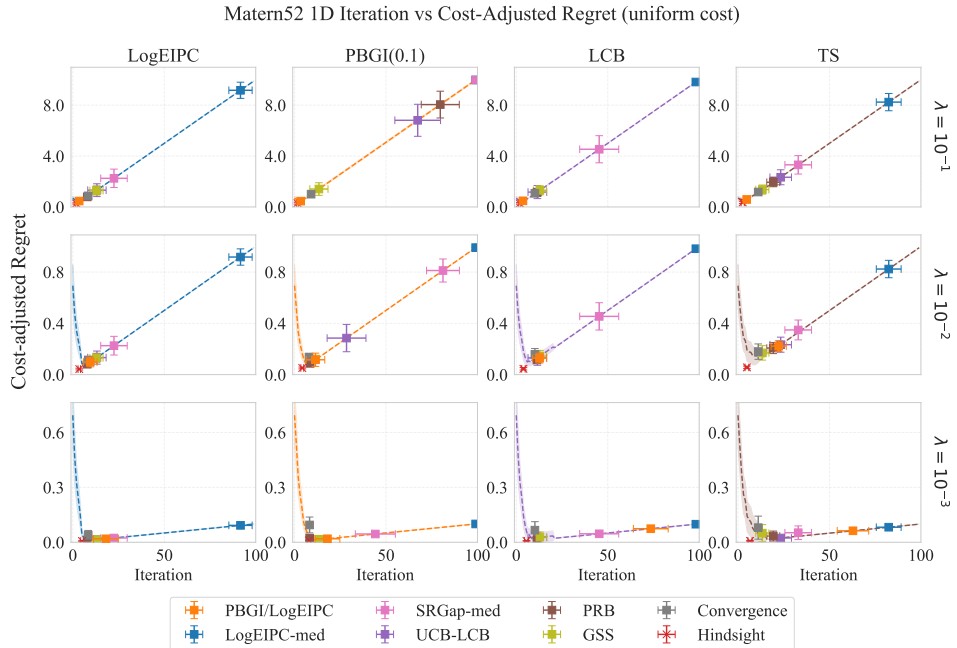

Figure 22: Comparison of cost-adjusted simple regret across acquisition function and stopping rule pairs in the 1D Bayesian regret experiments, with cost-scaling factor $\lambda = 10^{-1}, 10^{-2}, 10^{-3}$ and under uniform cost. The dashed line in each subplot represent fixed iteration stopping (e.g., the y-axis value of the line at iteration 50 represent the cost-adjusted regret when always stop at iteration 50).

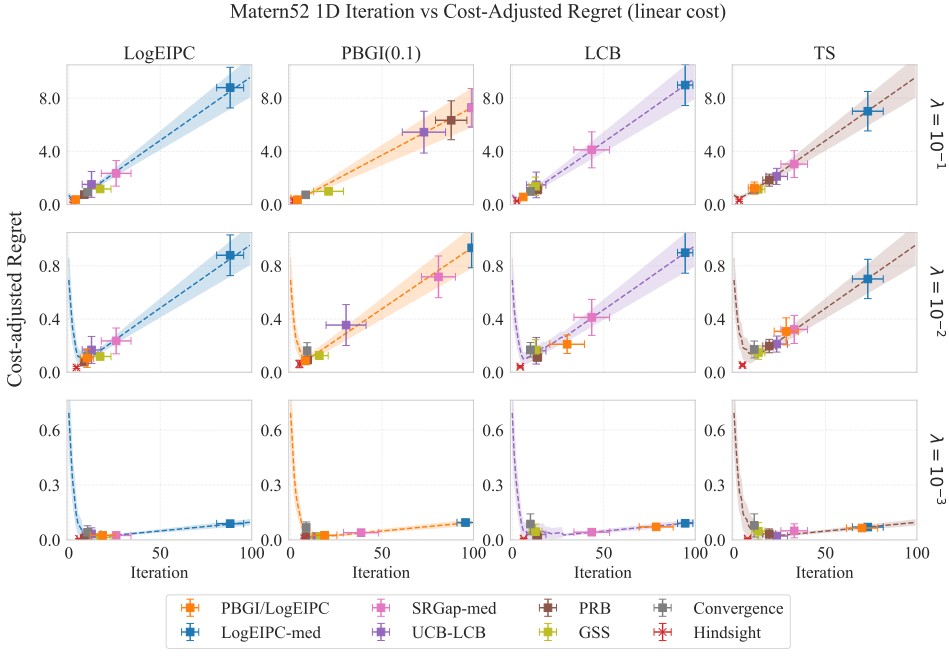

Figure 23: Comparison of cost-adjusted simple regret across acquisition function and stopping rule pairs in the 1D Bayesian regret experiments, with cost-scaling factor $\lambda = 10^{-1}, 10^{-2}, 10^{-3}$ and under linear cost. The dashed line in each subplot represent fixed iteration stopping (e.g., the y-axis value of the line at iteration 50 represent the cost-adjusted regret when always stop at iteration 50).

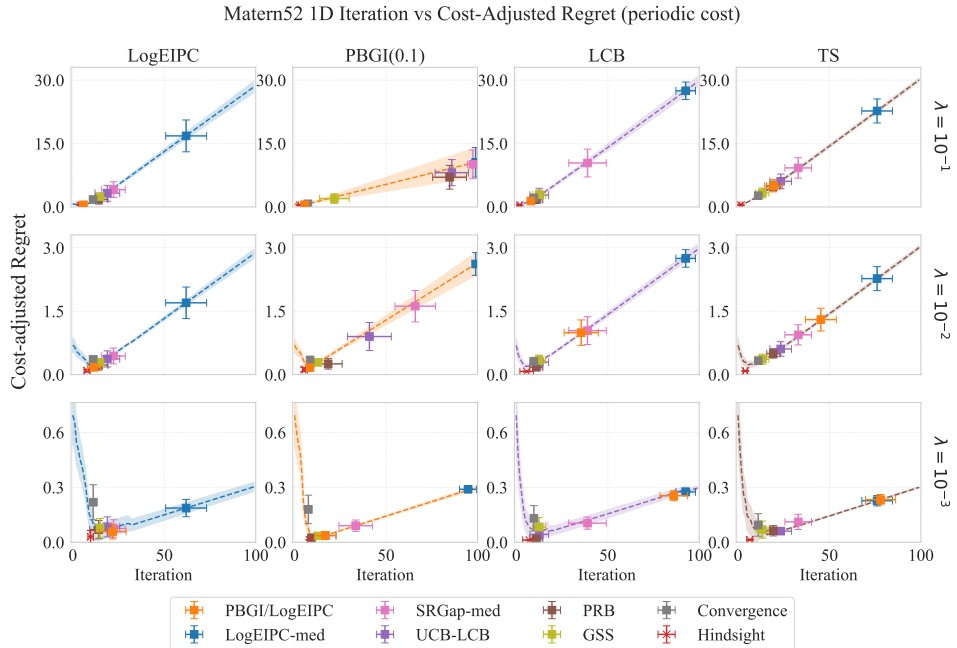

Figure 24: Comparison of cost-adjusted simple regret across acquisition function and stopping rule pairs in the 1D Bayesian regret experiments, with cost-scaling factor $\lambda = 10^{-1}, 10^{-2}, 10^{-3}$ and under periodic cost. The dashed line in each subplot represent fixed iteration stopping (e.g., the y-axis value of the line at iteration 50 represent the cost-adjusted regret when always stop at iteration 50).

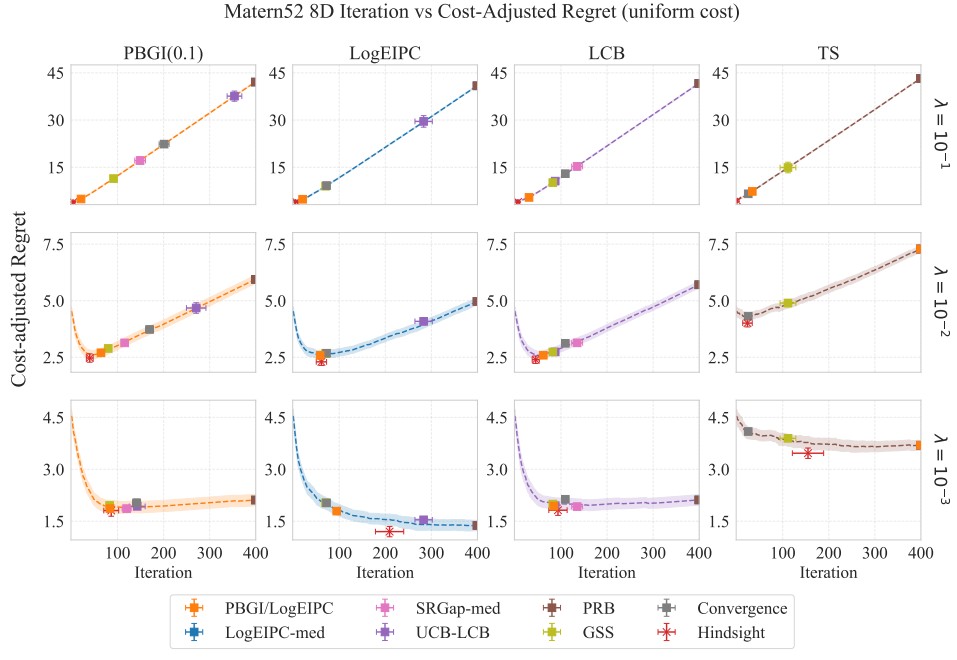

Figure 25: Comparison of cost-adjusted simple regret across acquisition function and stopping rule pairs in the 8D Bayesian regret experiments, with cost-scaling factor $\lambda = 10^{-1}, 10^{-2}, 10^{-3}$ and under uniform cost. The dashed line in each subplot represent fixed iteration stopping (e.g., the y-axis value of the line at iteration 50 represent the cost-adjusted regret when always stop at iteration 50).

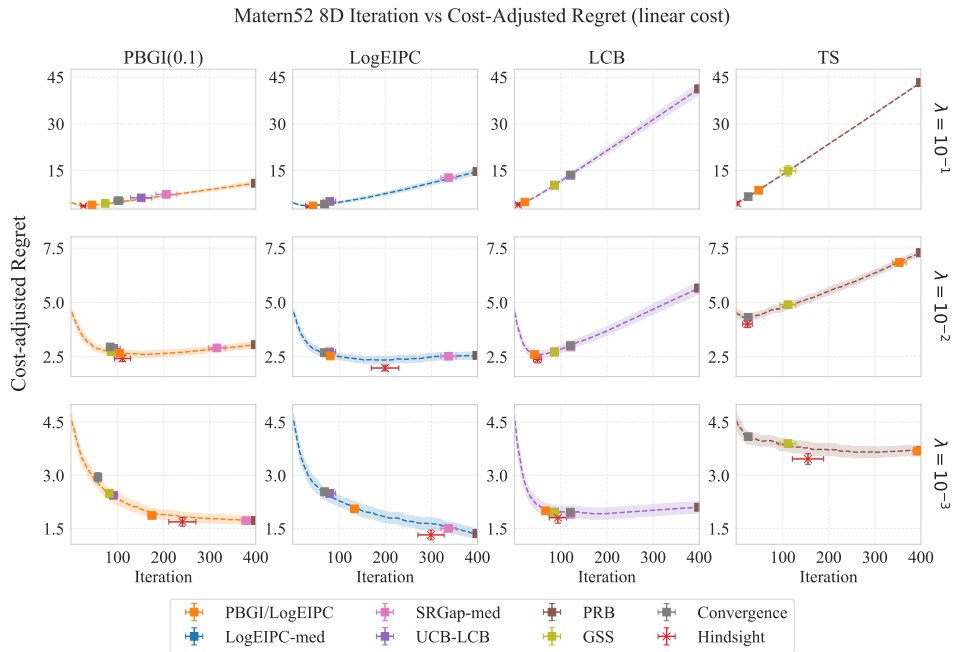

Figure 26: Comparison of cost-adjusted simple regret across acquisition function and stopping rule pairs in the 8D Bayesian regret experiments, with cost-scaling factor $\lambda = 10^{-1}, 10^{-2}, 10^{-3}$ and under linear cost. The dashed line in each subplot represent fixed iteration stopping (e.g., the y-axis value of the line at iteration $50$ represent the cost-adjusted regret when always stop at iteration $50$).

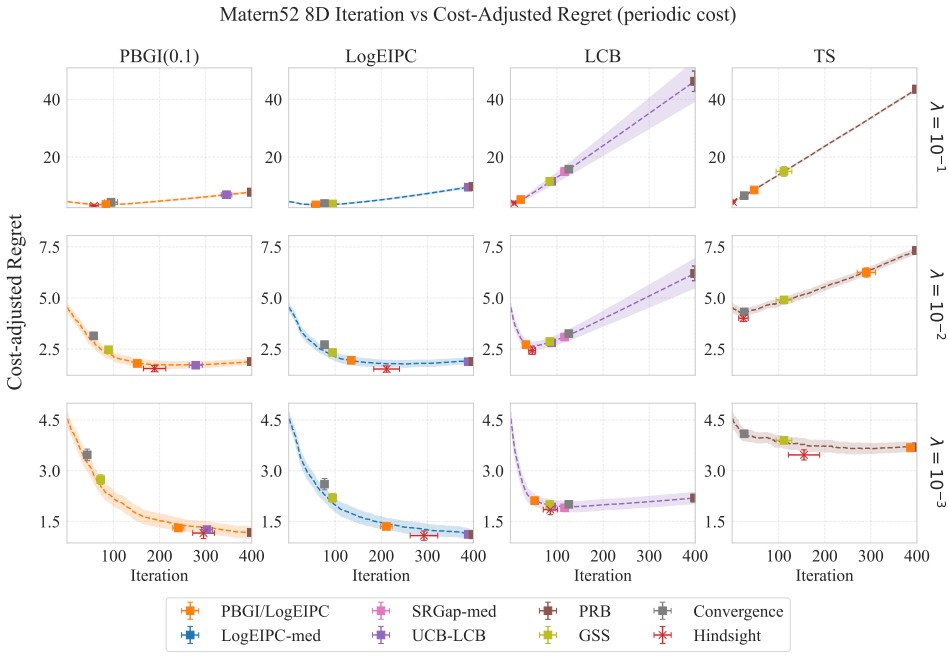

Figure 27: Comparison of cost-adjusted simple regret across acquisition function and stopping rule pairs in the 8D Bayesian regret experiments, with cost-scaling factor $\lambda = 10^{-1}, 10^{-2}, 10^{-3}$ and under periodic cost. The dashed line in each subplot represent fixed iteration stopping (e.g., the y-axis value of the line at iteration $50$ represent the cost-adjusted regret when always stop at iteration $50$).

