# OpenReview forum: "Cost-aware Stopping for Bayesian Optimization"
_ICLR.cc/2026/Conference — Submitted to ICLR 2026_

### Official Review · Reviewer_ypRo · 2025-10-23

**Soundness:** 3
**Presentation:** 3
**Contribution:** 3
**Rating:** 6
**Confidence:** 4

**Summary:**

This paper addresses the optimal stopping criterion problem—that is, determining when to stop the iterations in Bayesian optimization.
This is an extremely important issue in Bayesian optimization, whose main objective is to perform efficient exploration.
In particular, the paper makes a novel contribution by explicitly incorporating the evaluation cost of the unknown function into both the acquisition function and the corresponding stopping rule.

**Strengths:**

- It is useful that the paper provides a “conservative yet practically operable” guarantees.

- Although only a limited number of existing methods explicitly take evaluation costs into account, this paper discusses the combination of acquisition functions and stopping criteria and evaluates them experimentally, which adds meaningful insight to the field.

- It is also commendable that the paper provides a theoretically grounded method for scaling between cost and objective values (i.e., for setting the parameter \lambda).
This allows principled adjustment of the trade-off, rather than relying on purely heuristic tuning.

**Weaknesses:**

- There is no guarantee of termination within a finite number of iterations.

- The Bayesian optimality of the “index + stopping” rule in the Pandora’s Box problem relies on the assumptions of independence and discreteness, and this work cannot inherit the propety; the proposed method provides no strict guarantee of optimal stopping or regret rate in the Gaussian Process (GP) setting.

**Questions:**

Theorem 2 shows that, with the proposed stopping rule (combined with PBGI or LogEIPC), the expected cost-adjusted simple regret never exceeds that of the trivial baseline that stops immediately after the first evaluation. This provides a “safety-side” guarantee—one never wastes additional evaluations unnecessarily—which is novel among Bayesian-optimization stopping rules that explicitly incorporate cost. However, since the method does not guarantee finite-time stopping(which is not so a big requirement for a stopping criterion), it remains unclear how meaningful this no-worse guarantee is in practice. Without an assurance of eventual termination or bounds on stopping time, the result may be theoretically weak: it ensures safety in expectation but not operational reliability. In my view, the implications of Theorem 2 therefore require further discussion, as I am not yet fully convinced of its practical significance.

---

> ### Author Response · Authors · 2025-11-21
>
> We thank the reviewer for the valuable feedback and address the point below.
>
> ## Termination guarantee
> Thanks for raising this point. In the cost-aware Bayesian optimization setting, when the evaluation cost is negligible, non-termination is indeed possible and not necessarily undesirable. However, in most practical applications, evaluations incur a positive startup overhead, such as initialization time, resource allocation, or data loading. This implies a natural positive lower bound $c_0$, in which case our **Theorem 2 yields a finite-time termination guarantee**: with probability $1 - \delta$, the algorithm terminates in at most $\frac{U +C}{\delta \cdot c_0}$ iterations. We have included this result as a corollary in Section 3.1, with the proof provided in Appendix B.

---

> > ### Comment · Reviewer_ypRo · 2025-11-26
> >
> > I highly appreciate the importance of the stopping rule for BO and other AutoML related methods. With the additional result on the finite time stopping guarantee, I think this work is worth publishing at ICLR.
> > Could you please clarify or provide some explanation regarding the 2nd point of "Weaknesses" in my comment?
> > In a sense, the proposed method departs from the Bayesian optimality guaranteed under the conventional Pandora’s Box setting. I would like to reconfirm whether the proposed approach still provides, in any meaningful sense, an optimal stopping criterion or timing.

---

> ### Author Response · Authors · 2025-11-28
>
> ## On Bayesian Optimality and Theoretical Guaranties in the GP Setting
>
> We thank the reviewer for the positive comment and this important question. Bayesian optimality is indeed challenging in the correlated GP setting; even in the discrete Pandora’s Box model, introducing correlations already makes optimal stopping intractable. For this reason, fully carrying over the classical Bayesian-optimal ``index + stopping'' result is not currently feasible.
>
> That said, our stopping rule does have **meaningful theoretical guarantees**, and our experiments provide **empirical evidence** consistent with a stronger regret conjecture. We see potential for further guarantees, but developing them would likely require tools that go beyond what we can introduce in this paper. We outline these considerations in details below.

---

> ### Author Response · Authors · 2025-11-28
>
> **What we can guarantee.**
>
> In the GP setting, our stopping rule provides:
> (i) the ``**safety-side**'' guarantee noted by the reviewer—its expected cost-adjusted simple regret is never worse than immediate stopping; and
> (ii) a **finite-time termination** guarantee (under a mild positive lower bound on evaluation cost) requested by the reviewer.
>
>
> **What we believe is true (with empirical evidence) but cannot yet prove.**
>
> We conjecture that our stopping rule is competitive with the best *fixed* stopping time associated with the same acquisition function (LogEIPC or PBGI). Specifically, the performance gap between
> algorithm (A) using our PBGI/LogEIPC rule;
> algorithm (A') running the acquisition function for a fixed stopping time $T$,
> is bounded by a constant $O(c)$ that does not depend on $T$.
>
> Although we do not yet have the theoretical tools needed to establish this rigorously, we provide empirical comparisons against the best fixed $T$, and the differences are consistently small on synthetic benchmarks (Appendix D.4, Figure [20-25]), as well as empirical benchmarks without severe model-mismatch (Appendix D.4, Figure [12-17]). The dashed line in each of these figures represent fixed iteration stopping.
>
>
> **Why an optimality-like algorithm and proof are technically challenging.**
>
> Even in the discrete Pandora’s Box setting, introducing correlation makes optimal policies intractable, and existing theory (e.g. [1-3]) usually handles partially adaptive strategies with precommitted evaluation orders. In BO, precommitting on evaluation order is not consistent with how most acquisition functions behave (most are fully adaptive). Moreover, existing regret analyses for EI-type methods typically rely on RKHS-type worst-case assumptions (e.g. [4-6]) rather than Bayesian GP models, and to the best of our knowledge there is no established toolkit for analyzing regret that interacts cleanly with an adaptive stopping rule under GP correlation.
>
> **Why the conjecture may still be plausible.**
>
> All that said, there is some intuition suggesting the conjecture is plausible. The correlations induced by a GP prior are more structured than arbitrary correlations in the general correlated Pandora’s Box model. Under standard kernels, nearby points are either positively or negatively correlated, distant points are nearly independent, and the covariance structure does not encode pathological dependencies (e.g., ``the last bit of ($f(x)$) reveals ($f(y)$) exactly'').
>
> Our PBGI/LogEIPC rule stops only when the expected improvement at every location falls below the cost, so the marginal value of continuing is negative everywhere. If the cost-adjusted simple regret were to continue decreasing after this point, the selection process would effectively need to exhibit a form of superadditivity. Specifically, there should be a non trivial probability that given the correlation matrix and best seen value $y*$ at stopping, continuing to run BO algorithm $A$ for $l$ iterations yields greater expected cumulative improvement upon $y*$ than the sum of expected improvement at any fixed $l$ points. Given the structured and nonpathological nature of GP correlations, we expect any such superadditive interaction to be inherently bounded for adaptive algorithms. This is what informally supports the view that regret relative to fixed-time stopping should remain constant in scale.
>
>
> [1] Chawla, Shuchi, et al. "Pandora's box with correlations: Learning and approximation." 2020 IEEE 61st Annual Symposium on Foundations of Computer Science (FOCS). IEEE, 2020.
>
> [2] Chawla, Shuchi, et al. "Approximating Pandora's Box with Correlations." APPROX/RANDOM. 2023.
>
> [3] Gergatsouli, Evangelia, and Christos Tzamos. "Weitzman's rule for pandora's box with correlations." Advances in Neural Information Processing Systems 36 (2023): 12644-12664.
>
> [4] Nguyen, Vu, et al. "Regret for expected improvement over the best-observed value and stopping condition." Asian conference on machine learning. PMLR, 2017.
>
> [5] Gupta, Sunil, Santu Rana, and Svetha Venkatesh. "Regret bounds for expected improvement algorithms in Gaussian process bandit optimization." International Conference on Artificial Intelligence and Statistics. AISTATS, 2022.
>
> [6] Wang, Jingyi, et al. "Bayesian Optimization with Expected Improvement: No Regret and the Choice of Incumbent." arXiv preprint arXiv:2508.15674 (2025).

---

### Official Review · Reviewer_48nS · 2025-11-01

**Soundness:** 2
**Presentation:** 3
**Contribution:** 3
**Rating:** 4
**Confidence:** 4

**Summary:**

This work explores adaptive stopping of bayesian optimization where obtaining data points occurs a cost. The main contribution is a cost-aware stopping rule with theoretical guarantees.

**Strengths:**

* Adaptive stopping of Bayesian optimization is an important problem with practical relevance. This work is the first to study this problem in the cost-aware setting, which most closely resembles many practical settings.
* The proposed stopping rule comes with theoretical guarantees
* The work tackles a range of different cost-aware settings, such as budget-constrained and cost-per-sample, as well as settings where evaluation costs are or are not known in advance.
* A comprehensive set of baselines is considered, including "Hindsight", which provides a lower bound on achievable performance.
* Existing methodology and their relation to this work is well covered
* Clear structure, writing, and well-crafted figures.

**Weaknesses:**

* The number and type of benchmarks considered are limited. Additional and more different benchmarks would be helpful. This is my main issue with this work.
* A summary plot showing, e.g., average ranks or average normalized regret would be helpful for the overall evaluation. This would also allow you to display more different evaluation settings in the main paper.
* Unclear what effect does the debounce strategy ("requiring the stopping rule to consistently indicate stopping over several consecutive iterations before stopping optimization.") have? Would other approaches benefit from this, too?
* Code to reproduce experiments or an implementation of the algorithm is not provided. Some experimental details and the main hyperparameter settings are given.

**Questions:**

* How were the specific LCBench tasks chosen?
* Minor suggestion: Do not write "is defined in (5)", but "is defined in Equation (5)"

---

> ### Author Response · Authors · 2025-11-21
>
> We thank the reviewer for the valuable feedback and address the point below.
>
> ## Additional benchmarks
> Thanks for raising this point. We agree that broader benchmark coverage would further strengthen the empirical evaluation. We are working on additional benchmarks with varying evaluation costs, including the known-cost Pest Control, as well as the simulation-based Lunar Lander and Robotic Control considered in Xie et al. (2024). We will incorporate these results in the updated version of the manuscript.
>
> ## Summary plot
> Thanks for the helpful suggestion. We will include summary plots in the updated version of the manuscript.
>
> ## Effect of the debounce strategy
> Thanks for raising this point. The debounce strategy basically **smooths out the fluctuations** in the acquisition values. We have included an illustration of the effect in Figure 5 in the appendix, but we had forgotten to explicitly reference it in the main text; this has now been fixed.
>
> This smoothing is **not specific to our stopping rule**. In the updated Appendix C, we clarify that we apply the same debounce strategy to all acquisition-value–based stopping rules to mitigate fluctuations in their stopping signals (acquisition function values). For consistency, we also apply the 20-iteration moving-averaging to the non–acquisition-value–based GSS and Convergence stopping rules, although they do not require it.
>
> ## Availability of code and algorithm implementation
> The full implementation and code to reproduce all experiments are **included in the supplementary materials**.
> Specifically, we provide the complete source code for all acquisition functions and stopping rules, as well as scripts for reproducing every figure and table in the paper.
>
> ## LCBench task selection
> Thanks for raising this point. As noted in the manuscript, we used the first three tasks provided in the LCBench lite version. We are currently running additional LCBench tasks and will include the results later.
>
> ## Minor suggestion on equation cross-referencing
> Thanks for your suggestion! We've added "Equation" when cross-referencing every equation.

---

> ### Author Response · Authors · 2025-12-04
> **New LCBench Results and Summary Plot**
>
> We have now extended our experiments from the first three datasets in the LCBench six-dataset lite version to the **full 35 LCBench datasets**, and updated the per-dataset plots in the main text to summary plots showing performance aggregated over all tasks. Based on your valuable suggestions, we now report **min–max normalized cost-adjusted simple regret** in these summary plots and provide per-dataset cost-adjusted simple regret in the appendix. In the aggregated performance plots, our PBGI/LogEIPC stopping rule—when paired with either PBGI or LogEIPC—is consistently **among the top three** acquisition function--stopping rule pairs across three representative values of $\lambda$.
>
> On these 35 tasks, about **75\%** of them show our PBGI/LogEIPC stopping rule performing competitively when paired with either the PBGI or LogEIPC as the acquisition function. The remaining outlier tasks are almost all very small datasets (<10,000 instances), which might lead to severe misspecification. We believe this more comprehensive empirical evaluation supports the claim that our stopping rule is effective and robust in realistic, cost-aware AutoML settings.

---

### Official Review · Reviewer_XMVN · 2025-11-01

**Soundness:** 3
**Presentation:** 3
**Contribution:** 2
**Rating:** 4
**Confidence:** 3

**Summary:**

The paper proposes the PBGI/LogEIPC stopping rule for cost-aware Bayesian Optimization, which indicates when BO should be stopped to achieve the best trade-off between optimal values found and cost required. The proposed stopping rule is derived from the Pandora Box Gittins Index (PBGI) acquisition function. The authors also provide a connection between PBGI-based stopping rule with a stopping rule derived from Log Expected Improvement Per Cost (LogEIPC) acquisition function, demonstrating the equivalence of the two forms, hence the name PBGI/LogEIPC stopping rule. The stopping rule is theoretically guaranteed to maintain a bounded simple regret. The proposed stopping rule is evaluated against other stopping rules when applying to different acquisition functions (PBGI, EIPC, LCB and TS), on various synthetic and real-world benchmark problems.

**Strengths:**

-	The paper is well-written when explaining its methodology.
-	The work presents a stopping rule that is able to apply to a general BO algorithm.
-	There is a theoretical analysis to support the work.

**Weaknesses:**

1.	My primary concern lies in the novelty of the work. The concept of the proposed PBGI/LogEIPC stopping rule appears to be essentially identical to the PBGI acquisition function introduced by Xie et al. (2024). In particular, the stopping rule directly uses the same decision criterion that motivated the PBGI acquisition function—namely, determining whether to evaluate the new potential candidate point or to stop and accept the current best observation (see Sec. 3.2, Xie et al., 2024). This work largely reformulates this existing principle, expressing the same condition in the form of an explicit stopping rule rather than an acquisition function.
2.	Furthermore, the empirical results indicate that the proposed stopping rule performs well primarily when used together with PBGI, but not when combined with other acquisition functions (AFs). This outcome suggests that the rule is specifically tailored to the PBGI AF and does not generalize effectively to other AFs in BO. As such, the paper does not substantially extend the prior work, since Xie et al. (2024) already evaluated the PBGI framework, which implicitly captures this stopping behaviour as part of its policy. Therefore, the contribution of the current work appears incremental, mainly reiterating a property already discussed in the original PBGI study rather than introducing a broadly applicable stopping strategy.

**Questions:**

1.	As I might have missed some details, can the authors explain more about the novelty of this work?
2.	In the PBGI work, there are many acquisition functions to benchmark, e.g., KG, MES, MSEI, etc. Can the proposed stopping rule work with them? How was the performance?

---

> ### Author Response · Authors · 2025-11-21
>
> We thank the reviewer for the valuable feedback and address the point below.
>
> ## Novelty of our work
> Thanks for raising this point. The reviewer is correct that our stopping rule is motivated by the same design principle used to derive the PBGI acquisition function in Xie et al. (2024). However, the main novelty of our work does not lie in proposing a new design principle. Instead, our contributions lie in:
>
> (i) **New empirical evaluation of acquisition function -- stopping rules pairs**: Xie et al. (2024) evaluates PBGI only as an acquisition function under fixed-budget Bayesian optimization, and prior work about stopping rules (Makarova et al., 2022, Wilson, 2024) evaluates only stopping rules paired with one particular acquisition function. Our work provides the first systematic comparison across multiple acquisition function -- stopping rule pairs.
>
> (ii) **New theoretical guarantees** for the PBGI acquisition–stopping pair (also extended to EI and EIPC) under the correlated Bayesian optimization setting, which was only known to be Bayesian-optimal under the discrete independent Pandora's box setting before.
>
> (iii) **A new theoretical connection**: prior work has proposed heuristic stopping rules of the form “EI $\leq c$” (Nguyen et al. 2017) but not for EIPC and without principled justification. We show that the PBGI stopping rule (which is Bayesian-optimal under the independent discrete setting) is exactly equivalent to a stopping rule for EI(PC). This connection is not present in Xie et al. (2024).
>
> (iv) **Practical considerations that do not arise when using acquisition functions under fixed budgets**: e.g., whether stopping should be evaluated before or after posterior updates (we have added a discussion on this in our updated manuscript), and how to mitigate spurious stops due to imperfect GP model fitting or imperfect acquisition function optimization.
>
> To address the reviewer’s concern, we have also revised the final paragraph in the introduction to more accurately reflect these contributions and clarify the scope of our method.
>
> ## Generalizability of our stopping rule
> Thanks for bringing up this point.
> We agree that our proposed stopping rule is not designed to generalize to arbitrary acquisition functions.
> However, acquisition functions such as KG and MES are based on **fundamentally different design principles** (e.g., value-of-information or information-theoretic entropy reduction) and thus require their own matched stopping rules. Therefore, it is expected—and not a weakness of our method—that a PBGI-derived stopping rule does not pair effectively with acquisition functions derived from unrelated design principles.
>
> That said, this limitation does not diminish the contribution of our work. LogEI(PC) and PBGI currently represent **state-of-the-art methods for cost-aware Bayesian optimization**, and our goal is to extend them beyond the fixed-budget setting by providing a principled and practical adaptive stopping rule.
>
> We have added a discussion on such limitations to the final paragraph of the introduction in our updated manuscript.

---

### Official Review · Reviewer_Sjac · 2025-11-01

**Soundness:** 1
**Presentation:** 1
**Contribution:** 2
**Rating:** 2
**Confidence:** 4

**Summary:**

This paper proposes a stopping criterion for Bayesian optimization to enable efficient exploration in black-box function optimization problems where the evaluation cost varies across input points. The authors aim to suppress costly function evaluations while maintaining strong optimization performance, and they construct a new stopping rule based on existing acquisition functions used in cost-sensitive Bayesian optimization, PBGI and its equivalent LogEIPC. The proposed criterion adaptively determines whether to terminate the optimization process according to changes in the acquisition function value, thereby preventing unnecessary and expensive explorations in practice. Furthermore, numerical experiments on simulation and benchmark problems demonstrate that the proposed stopping method successfully avoids redundant high-cost evaluations while achieving optimization performance comparable to or better than existing approaches.

**Strengths:**

- This study is significant in that it clearly formalizes the previously ambiguous stopping condition in cost-sensitive Bayesian optimization, where the evaluation cost varies across input points. The proposed stopping criterion determines when to terminate the optimization based on the theoretical properties of the acquisition function, thereby preventing excessive exploration and unnecessary computational costs while achieving efficient and stable optimization.

- The proposed method is also notable for its application to AutoML tasks in a cloud environment, where its effectiveness is validated under practical conditions involving variable evaluation costs. Through these experiments, the study demonstrates that the proposed approach is beneficial for large-scale and high-cost real-world optimization problems, showing not only its theoretical contribution but also its high practical value.

**Weaknesses:**

- The meaning and derivation process of Equation (5) are not clearly explained in the text, making it extremely difficult to comprehend. Although the study relies heavily on the work of Xie et al., it lacks a self-contained explanation within the paper, which is problematic. Additional clarification should be provided so that readers can understand the content without referring to external sources.

- The theoretical analysis is based on an overly simplified assumption of a function with a constant mean, which does not adequately capture the complexity of real-world Bayesian optimization. This simplification undermines the generality and practical significance of the analytical results, creating a gap between the theoretical discussion and realistic applications.

- The proposed framework is highly dependent on the approach of Xie et al., and it shows limited applicability to acquisition functions other than PBGI and LogEIPC. Consequently, the generality and extensibility of the method are restricted, making it insufficient to claim its effectiveness across Bayesian optimization methods in general.

**Questions:**

- The definition and meaning of Equation (5) are extremely unclear. It is not evident whether \alpha_t^{\mathrm{PBGI}} represents the acquisition function itself or some other quantity. Moreover, the interpretation of the term involving (g) is insufficiently explained. In the upper condition part, the scales of EI and the cost c(x) appear to differ, making the interpretation of the entire equation ambiguous. Additionally, in the lower condition part, it is unclear whether f(x) denotes the objective function itself. If so, I completely do not understand why it can be used as acquisition function.

- The theoretical analysis in this paper focuses on a function with a constant mean, but this assumption seems far away from realistic Bayesian optimization settings. It should be clarified to what extent this simplified analysis holds practical meaning and validity for real-world optimization problems, and how the authors view the generalizability of their theoretical results.

- The proposed stopping criterion appears to be specifically designed for PBGI (or LogEIPC). It is unclear whether the insights and theoretical findings of this study can be applied, in some way, to other acquisition functions. If so, the applicable scope and required conditions should be explicitly discussed.

---

> ### Author Response · Authors · 2025-11-21
>
> We thank the reviewer for the valuable feedback and address the point below.
>
> ## Clarity of Equation (5)
> We thank the reviewer for pointing out Equation (5)'s lack of clarity. We agree that the two acquisition functions were not explained with sufficiently clarity.
> To address this, we first clarify that the notation of acquisition $\alpha_t$ was defined in earlier subsections (immediately before Section 2.1). In addition, we have **added brief one-sentence explanations** directly after the mathematical expressions for both acquisition functions in the revised manuscript.
>
>
> ## Constant-mean assumption in theoretical analysis
> We thank the reviewer for raising this point. We would like to clarify that assuming a constant prior mean is standard practice in GP–based Bayesian optimization, as in practical implementation one can always center the objective (subtract the mean function from the objective function) and work with a zero-mean prior. The goal of the constant-mean assumption in our initial presentation was only to simplify notation and highlight the key ideas of the analysis.
>
> Moreover, our theoretical results **do not rely on the mean being constant**. Our arguments extend directly to any prior mean function $\mu(\cdot)$, and the proofs require only minor notational modifications. To make this generality explicit, we have **updated the theorem statements** in Section 3.1 and proofs in Appendix B in the revised manuscript to allow an arbitrary prior mean function $\mu(\cdot)$.
>
> One subtlety arises in the quantities
> $U = \mu(x_1) - \mathbb{E}[\min_{x \in X} f(x)]$ and $C = c(x_1)$, which now depend on the choice of $x_1$. To obtain a tighter bound in our results (Theorem 2 and the new Corollary 3), one can pick the initial point $x_1$ as the one that minimizes $\mu(x_1)+c(x_1)$. We have included a remark on this in Appendix B.
>
>
> ## Generalizability of our stopping rule
> Thanks for bringing up this point.
> We agree that our proposed stopping rule is not designed to generalize to arbitrary acquisition functions.
> However, acquisition functions such as KG and MES are based on **fundamentally different design principles** (e.g., value-of-information or information-theoretic entropy reduction) and thus require their own matched stopping rules. Therefore, it is expected—and not a weakness of our method—that a PBGI-derived stopping rule does not pair effectively with acquisition functions derived from unrelated design principles.
>
> That said, this limitation does not diminish the contribution of our work. LogEI(PC) and PBGI currently represent **state-of-the-art methods for cost-aware Bayesian optimization**, and our goal is to extend them beyond the fixed-budget setting by providing a principled and practical adaptive stopping rule.
>
> We have added a discussion on such limitations to the final paragraph of the introduction in our updated manuscript.

---

> > ### Comment · Reviewer_Sjac · 2025-11-26
> >
> > Some aspects of the work became clearer through the authors’ response; however, my principal concerns remain unresolved. The paper still lacks sufficient definitions of equations, variables, and key components, preventing a proper assessment of the problem formulation, theoretical analysis, and general applicability. Moreover, the proposed framework appears to depend heavily on the approach of Xie et al., which restricts its generality and extensibility, making the current claims about broader effectiveness in Bayesian optimization insufficiently supported. For these reasons, my evaluation remains unchanged, and I will keep the original score.

---

> ### Author Response · Authors · 2025-11-26
>
> We appreciate the reviewer’s follow-up. While we understand the reviewer’s concerns, we respectfully disagree with the expectation that a universal stopping rule should exist across acquisition functions built on fundamentally different principles, especially since some of these acquisition functions are not cost-aware. Our stopping rule shows for the first time that state of the art acquisition functions PBGI and LogEIPC admit the same stopping criterion, which is already surprising.
>
> Regarding clarity, we have already revised notation and definitions according to the reviewer’s suggestion. If there are still specific expressions or variables the reviewer finds unclear, we would welcome a pointer so we can address it directly in future versions.

---

### Author Response · Authors · 2025-12-04

Dear Area Chair,

Thank you for handling this submission under the unusual circumstances of this year’s rebuttal period! Below we briefly summarize (i) what the reviewers agree are our main strengths, (ii) the reviewers’ key concerns and how we addressed them during the rebuttal phase, and (iii) new experiments that we were not able to report back before the pause.

## What the reviewers agree
Reviewers acknowledge that adaptive stopping in cost-aware Bayesian optimization is practically important (**48nS**, **Sjac**), that our stopping rule provides theoretically grounded and practically usable guarantees (**XMVN**, **48nS**, **ypRo**), and that the paper offers a clear, well-structured empirical study of acquisition–stopping pairs across multiple cost-aware settings with comprehensive baselines and good coverage of related work (**48nS**, **ypRo**).

## How we addressed the main concerns in the rebuttal phase
Reviewer **Sjac**’s main concern is that they were a bit confused by our notations, and they were concerned that our constant-mean GP prior assumption is an oversimplification. In response, we tightened notation, and added explicit one-sentence explanations immediately after the PBGI and LogEIPC formulas. Importantly, our theoretical results do not rely on the constant-mean GP assumption. We updated the theorem statements and the proof to make this explicit.

Reviewer **XMVN**’s main concern is that our contribution might be seen as a reformulation of PBGI with limited novelty and scope. In response, we clarified the key novel aspects of our work, which go well beyond reformulation. Specifically, we provide
(a) a novel theoretical connection between PBGI and EIPC stopping rule that, to our knowledge, has not been established in prior work; (b) new theoretical guarantees for the PBGI acquisition–stopping pair (also extended to EI and EIPC) which was only known in much more limited setting before; (c) extensive empirical evaluation of acquisition function--stopping rules pair v.s. the fixed time stopping in prior work.

Reviewer **48nS**’s main concern is that the empirical evaluation would be stronger with more benchmarks and summary plots. Unfortunately due to the pause in the rebuttal period, we are not able to further engage with the reviewer with our additional LCBench results. These new results are updated in our revised manuscript and summarized here in **New LCBench results**.

Reviewer **ypRo**’s main concern is the practical meaning of Theorem 2 without a finite-time stopping guarantee; in response, we added a corollary of Theorem 2 explicitly showing termination guarantee under a mild positive lower bound $c_0 > 0$ on evaluation cost. After our update, the reviewer was satisfied and explicitly said *I think this work is worth publishing at ICLR*. We were also engaging in an interesting discussion with Reviewer ypRo on the plausibility and difficulty of extending the theoretical optimality of PBGI discrete independent setting to correlated GP, but it was unfortunately cut short by the pause.

## New LCBench results (Reviewer 48nS)
We have now extended our experiments from the first three dataset in the LCBench six-dataset lite version to the **full 35 LCBench datasets**, and updated the main figures to show performance aggregated over all tasks.

On these 35 tasks, about **75\%** of them show our PBGI/LogEIPC stopping rule performing competitively when paired with either the PBGI or LogEIPC as the acquisition function. The remaining outlier tasks are almost all very small datasets (<10000 instances), which might lead to severe misspecification. This larger benchmark set supports the reviewers’ view that our stopping rule is effective and robust in realistic, cost-aware AutoML settings.

## Summary
In summary, most reviewers recognize the **importance of the automatic stopping problem** (**Sjac**, **48nS**, **ypRo**) and see clear value in our theoretical and practical contributions (**XMVN**, **48nS**, **ypRo**). Further, their **main technical concerns have been addressed** in the revised manuscript (as described above), and the **new LCBench results further strengthen our empirical case**. We hope this helps you in your decision.

---

### Meta-Review · Area_Chair_ULrx · 2025-12-09

**Summary:**

This paper propose a cost-aware stopping rule for Bayesian optimization that are theoretically grounded, free of heuristic tuning, and consistently achieve competitive cost-adjusted simple regret on hyperparameter optimization tasks.

### Pros

*  A theoretical grounding paper
* The paper addresses a practical and significant problem
* well-written and easy to follow

### Cons

* Limited generalizability (critical issue)
* Insufficient empirical validation
* Theoretical limitations

### AC's evalution

1. from reviews and rebuttals

This paper receives 6442. It faces strong opposition from three out of four reviewers. Reviewer Sjac (2) remains the strongest critic. They explicitly stated that their "principal concerns remain unresolved," particularly regarding the method's lack of generalizability and poor clarity. Reviewer 48nS (4) identified the "limited benchmarks" as a fatal flaw in the initial submission. The massive addition of experiments during rebuttal often signals that the paper was not ready for submission, and this reviewer remains unconvinced by the rushed update.
Reviewer XMVN (4) aligns with the opposing camp. Their negative stance reinforces the view that the paper's contribution is too narrow or theoretically fragile to warrant acceptance at ICLR.
Reviewer ypRo (6): The sole supporter, finding the theoretical insight valuable. However, their support is not strong enough to override the consensus of the other three reviewers.




2. from AC's reading

I recommend Rejection. With three reviewers (Sjac, 48nS, XMVN) arguing against acceptance, it is clear. The key technical bottleneck that the stopping rule is hard-coded to a specific acquisition function limits its impact significantly. Combined with the clarity issues pointed out by Sjac and the initial experimental weakness noted by 48nS, the paper does not meet the high bar for ICLR. The unresolved status of major concerns after the rebuttal confirms that another round of revision is necessary.

**Reviewer Concerns:**

Resolved Concerns:

1. Constant Mean Assumption: The authors clarified the extension to dynamic means, which was technically accepted but didn't solve the broader applicability issue.

Outstanding Concerns:

1. Generalizability (Reviewer Sjac & XMVN): The method's inability to support standard acquisition functions (like EI or KG) is a critical flaw. The authors' argument that "different principles need different rules" was rejected by the reviewers, who expect a more unified solution.

2. Experimental Rigor (Reviewer 48nS): The initial lack of comprehensive benchmarks created a deficit of trust that the rebuttal failed to fully recover.

3. Clarity (Reviewer Sjac): Persistent issues with definitions and mathematical clarity make the paper difficult to verify and consume.

**Reviewer Scores:**

Two of them participated discussions

Reviewer Sjac maintain 2. He expressed dissatisfaction with the Rebuttal, particularly regarding the generality of the methods and the clarity of definitions, explicitly stating “Evaluation remains unchanged.”

Reviewer 48nS will maintain 4. While acknowledging the authors' supplementary experiments, his initial impression of the experimental weakness was strong, and his overall enthusiasm for the paper was low. This typically leans toward rejection when significant flaws exist.

Reviewer XMVN will maintain 4. Opposes the paper, likely sharing concerns about utility or theory.

Reviewer ypRo keep 6 . The only positive voice, but stands alone.

---

### Decision · Program_Chairs · 2026-01-26

Reject